# Archaeal lipid biomarker constraints on the Paleocene-Eocene carbon isotope excursion

Felix J. Elling [1]*, Julia Gottschalk [2], Katiana D. Doeana[1], Stephanie Kusch [1,3], Sarah J. Hurley[1,4] & Ann Pearson[1]

A negative carbon isotope excursion recorded in terrestrial and marine archives reflects massive carbon emissions into the exogenic carbon reservoir during the Paleocene-Eocene Thermal Maximum. Yet, discrepancies in carbon isotope excursion estimates from different sample types lead to substantial uncertainties in the source, scale, and timing of carbon emissions. Here we show that membrane lipids of marine planktonic archaea reliably record both the carbon isotope excursion and surface ocean warming during the Paleocene-Eocene Thermal Maximum. Novel records of the isotopic composition of crenarchaeol constrain the global carbon isotope excursion magnitude to −4.0 ± 0.4‰, consistent with emission of >3000 Pg C from methane hydrate dissociation or >4400 Pg C for scenarios involving emissions from geothermal heating or oxidation of sedimentary organic matter. A pre-onset excursion in the isotopic composition of crenarchaeol and ocean temperature highlights the susceptibility of the late Paleocene carbon cycle to perturbations and suggests that climate instability preceded the Paleocene-Eocene Thermal Maximum.

[1] Department of Earth and Planetary Sciences, Harvard University, Cambridge, MA 02138, USA. [2] Lamont-Doherty Earth Observatory, Columbia University of the City of New York, Palisades, NY 10964, USA. [3]Present address: CologneAMS, University of Cologne, Bernhard-Feilchenfeld-Str. 11, 50969 Cologne, Germany. [4]Present address: Department of Geosciences, University of Colorado, Boulder, CO 80309, USA. *email: felix_elling@fas.harvard.edu

The early Paleogene witnessed frequent intervals of transient warming, so-called hyperthermals, superimposed on a greenhouse background climate[1]. The most intense hyperthermal, the Paleocene-Eocene Thermal Maximum (PETM), was a ~200 ka global climate perturbation[2] associated with increased atmospheric $CO_2$ levels[3,4], ocean warming[5], and ocean acidification[6]. Widespread observations of a negative carbon isotopic excursion (CIE) in various geological substrates indicate substantial emissions of $^{13}C$-depleted carbon into the biosphere-ocean-atmosphere (exogenic) reservoir during the PETM[7–9]. Due to the fossil nature of the carbon and the potentially rapid release, the PETM represents a close albeit imperfect analog for modern anthropogenic climate change[10–12]. Accurately reconstructing the amount, origin, and timing of carbon emissions during the PETM[13,14] is a critical step towards understanding the associated climatic forcing and feedback mechanisms, and thus the impacts of carbon cycle perturbations on the past and future Earth system[15].

The duration and magnitude of the CIE are important constraints on the amount and source of the emitted carbon[8,16]. The magnitude of the CIE can in principle be reconstructed through $\delta^{13}C$ measurements of carbon-bearing substrates in geologic records, and is thus commonly used for mass balance calculations of PETM emission[8,9,16] (compiled in Supplementary Data 1) and as a tuning target in simulated emission scenarios[11,13,17,18]. Despite more than 160 published CIE estimates from various sample substrates[16], the true magnitude of the CIE of the exogenic carbon reservoir remains elusive. In particular, it remains unclear whether the CIE reconstructed from a given sample substrate is representative of the CIE of the entire exogenic carbon reservoir, since equilibration between the exogenic sub-reservoirs may depend on emission rates and size, carbon isotopic composition and location of the source, and oceanographic changes[19]. The CIEs recorded in different sample substrates vary from −0.6 to −5‰ in marine carbonates to −1 to −8‰ in bulk organic carbon from both marine and terrestrial records[16]. The wide range of CIE values has led to a diversity of PETM carbon emission scenarios ranging from 2000–20,000 Pg C (refs. [16,17] Supplementary Data 1). These scenarios invoke a wide array of possible carbon sources, including enhanced methane emissions from hydrate dissociation[8], geothermal heating of organic carbon[20], and oxidation of permafrost or other generic organic carbon sources[9,21]. Carbon cycle modeling based on observed changes in ocean carbonate saturation and ocean pH yield CIE-independent, but diverging estimates of ~3000–7000 Pg C (refs. [13,14]) and > 10,000 Pg C (ref. [17]), respectively. However, reconstructions of the isotopic composition of the carbon source from these models still depend on assumptions about the CIE magnitude[18].

The discrepancy in the magnitude of the CIE between sample substrates and across sampling locations likely originates from changes in geochemical and biological factors across the PETM that influence $\delta^{13}C$ values of the substrate. These factors include diagenetic overprints, source heterogeneity, and vital effects of the source organisms[16,19,22]. For instance, the carbon isotopic composition of soil bulk organic matter is affected by variations in higher plant assemblages[23] as well as by variable fractionation imparted by changes in atmospheric $CO_2$ concentrations and associated growth rates[3]. These factors similarly affect the carbon isotopic composition of marine phytoplankton biomass[24] and thus bias reconstructions from bulk sediment organic matter (often towards larger CIEs), in addition to offsets resulting from varying admixtures of terrigenous organic matter into marine sediments[19]. By contrast, $\delta^{13}C$ records of bulk carbonate and foraminifera may underestimate the magnitude of the CIE due to biases imposed by carbonate dissolution[25], diagenesis[26], and vital effects[27]. Collectively, these uncertainties confound not only the quantification of carbon emissions during the PETM[16], but also our understanding of temporal release patterns[11,17,28], climate sensitivity[15], and ecosystem recovery[10].

A proxy for the CIE that is insensitive to mineral dissolution and has negligible or readily quantifiable vital effects would help constrain carbon emissions during hyperthermals. The archaeal lipid biomarker crenarchaeol may satisfy these criteria for the marine environment of the Paleogene. This organic compound originates from Thaumarchaeota[29], a group of marine planktonic archaea that fix dissolved inorganic carbon (DIC)[30]. $\delta^{13}C_{cren}$ should thus primarily reflect $\delta^{13}C_{DIC}$ at the habitat depth of Thaumarcheaota[31,32], which predominantly reside at the base of the photic and upper sub-photic zones[33,34]. Analysis of thaumarchaeal lipids additionally provides coupled ocean temperature estimates through the $TEX_{86}$ proxy[35], such that both $\delta^{13}C_{DIC}$ and ocean temperature can be obtained on the same substrate, circumventing stratigraphic uncertainties inherent to CIE–temperature correlations based on different substrates.

Here we apply spooling-wire microcombustion–isotope ratio mass spectrometry of $\delta^{13}C_{cren}$ (described in detail in the Methods section and in ref. [32]) to reconstruct the PETM CIE in thaumarchaeal lipids from three globally distributed sites. Values of $\delta^{13}C_{cren}$ are then compared to the CIE recorded in newly generated and previously published $\delta^{13}C$ records of total organic carbon (TOC)[36], benthic foraminifera[37], dinocysts[19,38], and $n$-alkanes[39]. We argue that under PETM boundary conditions (pH ~7.5, ref. [6]; [DIC] ≥ 2 mM) $\delta^{13}C_{cren}$ is offset by a predictable fractionation factor from $\delta^{13}C_{DIC}$ (refs. [40,41]) and that the change in $\delta^{13}C_{cren}$ during the PETM closely approximates the CIE in the global exogenic carbon reservoir. We further couple $\delta^{13}C_{cren}$ to $TEX_{86}$ analyses which allows simultaneous reconstruction of CIE and temperature change in low-carbonate sediments, thereby circumventing confounding effects of carbonate dissolution prevalent during the PETM and other climate warming events. Our independent CIE estimate derived from $\delta^{13}C_{cren}$ represents an important step towards constraining PETM carbon emissions.

## Results

**Study area.** We sampled the Paleocene-Eocene boundary in three marine sediment cores (Fig. 1) retrieved during expeditions ODP 174AX (Ancora site, Hole A/B, New Jersey shelf), ODP 189 (Hole 1172D, core 15R section 4–5, Tasman Sea), and IODP 302 (Hole M0004A, cores 28X-32X, Arctic Ocean). During the Paleocene and Eocene, the New Jersey shelf and Tasman Sea sites were located on continental margins of the north-west Atlantic and south-west Pacific Oceans, respectively[36,42]. The Arctic Ocean site was located on the submerged northern flank of the Lomonosov Ridge[43]. Records of $\delta^{13}C_{cren}$ were generated for all three sites and were compared against $\delta^{13}C$ changes in other substrates at these sites. Newly generated (New Jersey) and existing (Arctic Ocean[43], Tasman Sea[36]) $TEX_{86}$ records were used to compare changes in ocean temperature and $\delta^{13}C_{cren}$ across the PETM. To assess the potential influence of terrigenous input of crenarchaeol on $\delta^{13}C_{cren}$, records of BIT (branched-over-isoprenoid tetraether index), a semi-quantitative tracer for input of soil organic matter[44], were generated for the New Jersey core, while literature data were used for the Tasman Sea[36] and Arctic Ocean[43] cores. $\delta^{13}C_{TOC}$ records were generated for the Ancora and Arctic Ocean cores and literature data were used for the Tasman Sea site[36].

**Crenarchaeol stable carbon isotope records.** Our new $\delta^{13}C_{cren}$ data allow an assessment of the magnitude of the crenarchaeol CIE ($CIE_{cren}$) at each location. For the Ancora and Tasman Sea datasets, we estimate $CIE_{cren}$ as the difference between mean

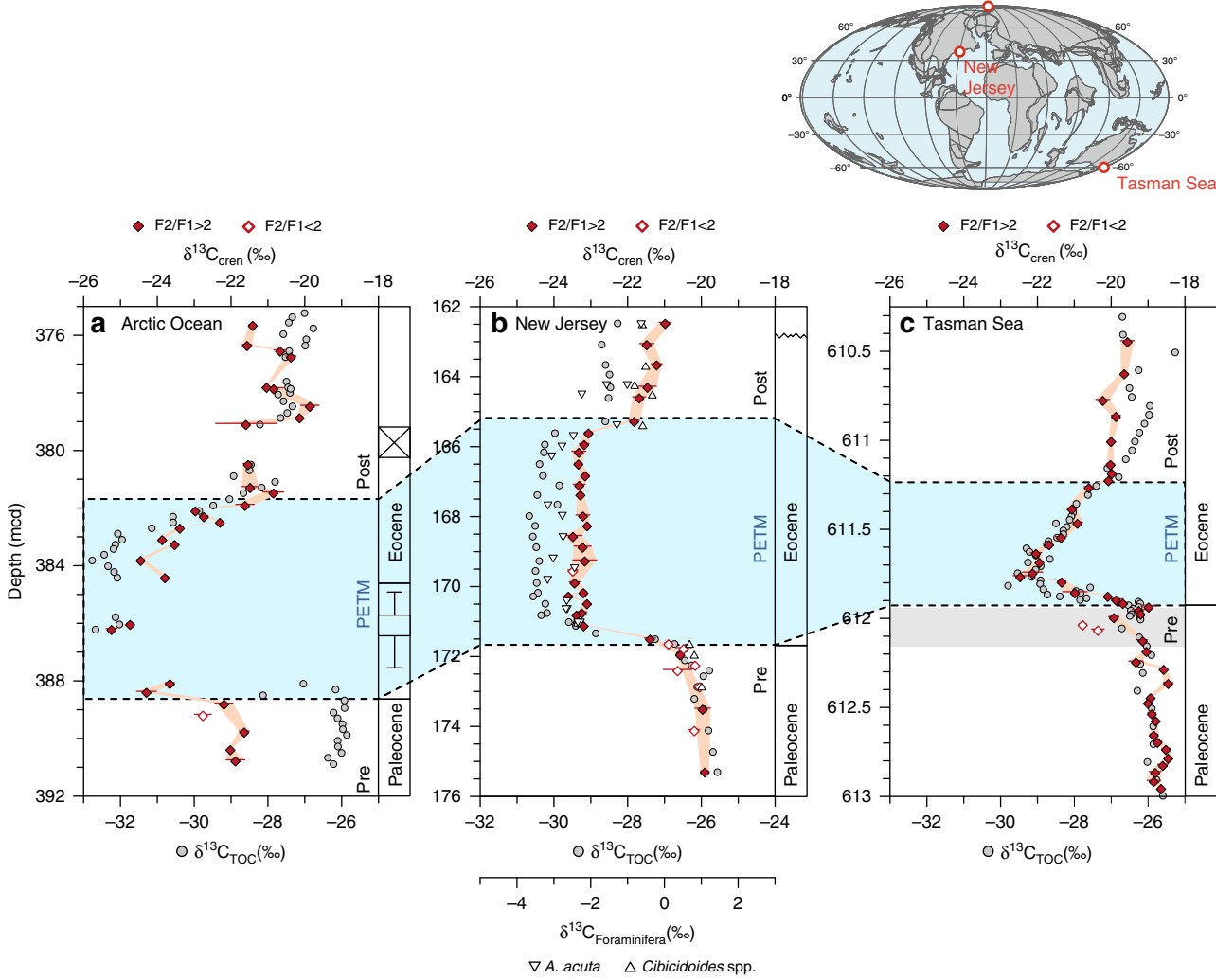

**Fig. 1** The Paleocene-Eocene carbon isotope excursion recorded in crenarchaeol and other marine substrates. Carbon isotopic ratios of crenarchaeol ($\delta^{13}C_{cren}$), total organic carbon ($\delta^{13}C_{TOC}$; Arctic Ocean, New Jersey: this study; Tasman Sea: ref. [36]) and the foraminifera *Anomalinoides acutus* and *Cibicidoides* spp. ($\delta^{13}C_{Foraminifera}$; ref. [37]) at three globally distributed sites (**a**, Arctic Ocean; **b**, New Jersey shelf; **c**, Tasman Sea). Foraminifera are preserved only in the New Jersey shelf record. Closed and open symbols for $\delta^{13}C_{cren}$ represent samples with high and low signal-to-background ratios (F2/F1; see Methods), respectively. Error bars of $\delta^{13}C_{cren}$ represent one standard deviation of replicate analyses ($n = 5$). Error bars of $\delta^{13}C_{cren}$ are not displayed where they are smaller than the symbol. Boundaries of the PETM and pre-onset event are shown in blue (based on refs. [36,42,43]) and gray shading, respectively. Error bars in the stratigraphic column of the Arctic Ocean record indicate uncertainty in sample position due to poor core recovery. Inset map (**d**) shows sampling locations and plate configuration (reconstructed using the Ocean Drilling Stratigraphic Network webservice, http://www.odsn.de/ and the model of ref. [77]) during the late Paleocene (ca. 55 Ma)

pre-PETM and mean peak-PETM values (with the $CIE_{cren}$ error representing the propagated uncertainties, i.e., 1σ standard deviations of the means)[16]. The intervals of the pre- and peak-PETM were determined through ramp function fitting[45] (rampfit; Supplementary Fig. 1). For the Arctic Ocean record, owing to the coring gaps, we report the CIE as the difference between the pre-PETM $\delta^{13}C_{cren}$ mean and the lowest $\delta^{13}C_{cren}$ value during the PETM. We find a $CIE_{cren}$ of $-3.12 \pm 0.20$‰ for the Ancora site and $-3.28 \pm 0.21$‰ for the Tasman Sea (Fig. 1). The $CIE_{cren}$ estimate for the Arctic Ocean is $-3.31 \pm 0.23$‰. These estimates are all consistent within 1σ uncertainties, with an average $CIE_{cren}$ of $-3.24 \pm 0.37$‰.

Following our rampfit approach, the CIE in total organic carbon (TOC) is $-4.31 \pm 0.39$‰ at Ancora, $-3.06 \pm 0.37$‰ in the Tasman Sea (data from ref. [36]), and $-6.67 \pm 0.22$‰ in the Arctic Ocean. These values are significantly more heterogeneous than our $\delta^{13}C_{cren}$-based estimates, possibly due to

variable proportions of terrigenous and marine input influencing the $\delta^{13}C_{TOC}$ signal, which leads to an overestimation of the CIE at least for the Arctic Ocean record[19]. At the New Jersey shelf, the only site studied here with sufficiently preserved carbonates, benthic foraminifera[37,46] indicate a CIE of $-3.54 \pm 0.34$‰ at Ancora and $-3.76 \pm 0.44$‰ at Bass River (rampfit approach; Supplementary Fig. 1).

**Temperature records and provenance of archaeal lipids**. Low BIT indices of ~0.3 indicate a marine provenance of glycerol dibiphytanyl glycerol tetraethers (GDGTs) throughout the Ancora record, which is similar to previous records from Bass River[38] and the Tasman Sea[43] (Supplementary Fig. 2). In contrast, BIT values for the Arctic Ocean core[43] indicate significant terrigenous input before the CIE, with a drop in BIT during the CIE indicating decreased relative contributions from terrigenous GDGTs during the PETM.

At the depth of the CIE, TEX$_{86}$ values increase by ~0.2 in the Ancora record, indicating ocean warming, and return to pre-CIE levels after the PETM (Fig. 2). A similar pattern is observed in the Bass River[38] and Tasman Sea records[36] (Fig. 2). By contrast, TEX$_{86}$ values do not increase significantly over the course of the CIE in the Arctic Ocean record (Supplementary Fig. 3; ref. [43]), which has previously been attributed to unusually high abundances of tricyclic GDGT at this site relative to modern core-top compilations[43]. However, the modified TEX$_{86}$' index, which omits tricyclic GDGT but is still well-correlated to SST in modern core-top sediments[43], shows an increase of 0.07 in parallel with the CIE, and returns to pre-CIE levels after the PETM (Fig. 2; data from ref. [46]). For comparison, TEX$_{86}$' values calculated for the Ancora and Tasman Sea records also increase in parallel with TEX$_{86}$ (Supplementary Fig. 3). There is no consistent depth offset between changes in the TEX$_{86}$ (or TEX$_{86}$') and δ$^{13}$C$_{cren}$ signals at the PETM onset, potentially indicating synchronous change in temperature and CIE. However, incomplete recovery in the Arctic Ocean record[39,43] and potential condensation in the Tasman Sea record[47] could obscure potential lead-lag relationships.

## Discussion

The distinctive biology of Thaumarchaeota yields important constraints on the carbon isotope systematics of their lipids. Thaumarchaeota fix carbon using the 3-hydroxypropionate/4-hydroxybutyrate pathway exclusively from bicarbonate ($HCO_3^-$) rather than from $CO_2$ (ref. [30]). Flux balance modeling[41] and environmental data[40] suggest that carbon isotopic fractionation in marine Thaumarchaeota originates from passive $CO_2$ transport into the cell combined with slow conversion to $HCO_3^-$, resulting in a dependence of the expressed fractionation factor on the dissolved $CO_2$ concentration. Importantly, the effect is inverse to the influence of $pCO_2$ on carbon isotopic fractionation in

phytoplankton, resulting in a smaller fractionation factor ($\varepsilon_{DIC-Cren}$) at high dissolved $CO_2$ concentrations[40,41]. Due to the $pCO_2$ increase during the PETM $\varepsilon_{DIC-Cren}$ may have been smaller by $0.75 \pm 0.15$‰ (range 0.6–0.9‰) during the peak-PETM compared to the pre-PETM (assuming Paleocene-Eocene atmospheric $pCO_2$ of ~850–2200 ppm, refs. [13,17] Supplementary Fig. 4). Accounting for this effect leads to a δ$^{13}$C$_{cren}$-based CIE in DIC (CIE$_{DIC}$) of $-3.87 \pm 0.25$‰ at Ancora, $-4.03 \pm 0.27$‰ in the Tasman Sea, $-4.06 \pm 0.27$‰ in the Arctic Ocean, yielding an average of $-4.0 \pm 0.4$‰. Tighter constraints on Paleocene-Eocene $pCO_2$ will allow reducing the uncertainty in $\varepsilon_{DIC-Cren}$ estimates. Regardless, we expect the $pCO_2$-dependent sensitivity of $\varepsilon_{DIC-Cren}$ to be relatively constant across archaeal species, as cell morphology and physiology are highly similar among marine planktonic Thaumarchaeota[48–50] and the 3-hydroxypropionate/4-hydroxybutyrate pathway is phylogenetically conserved[30]. Therefore, minimal effects are expected from changes in the diversity or distribution of thaumarchaeal clades between the modern ocean and past ecosystems. Studies with additional thaumarchaeal cultures will be needed to fully assess potential impacts of changes in community composition on $\varepsilon_{DIC-Cren}$ and δ$^{13}$C$_{cren}$ records.

Due to their ecological niche as chemolithoautotrophic ammonia oxidizers, Thaumarchaeota are most abundant in a narrow interval at the base of the photic zone[33,34]. Consequently, thaumarchaeal lipids appear to be predominantly sourced from this zone[51], despite the presence of Thaumarchaeota throughout the ocean water column[52]. Contributions from meso- and bathypelagic Thaumarchaeota should therefore not be a major confounding factor for sedimentary δ$^{13}$C$_{cren}$ records, and the signal should principally reflect basal photic zone δ$^{13}$C$_{DIC}$. A point-source origin of δ$^{13}$C$_{cren}$ from the basal photic zone is further supported by δ$^{13}$C$_{cren}$ records of Pleistocene Mediterranean sapropel events S3 and S5 (ref. [53]), where significant changes in water column hydrography and changing contributions from

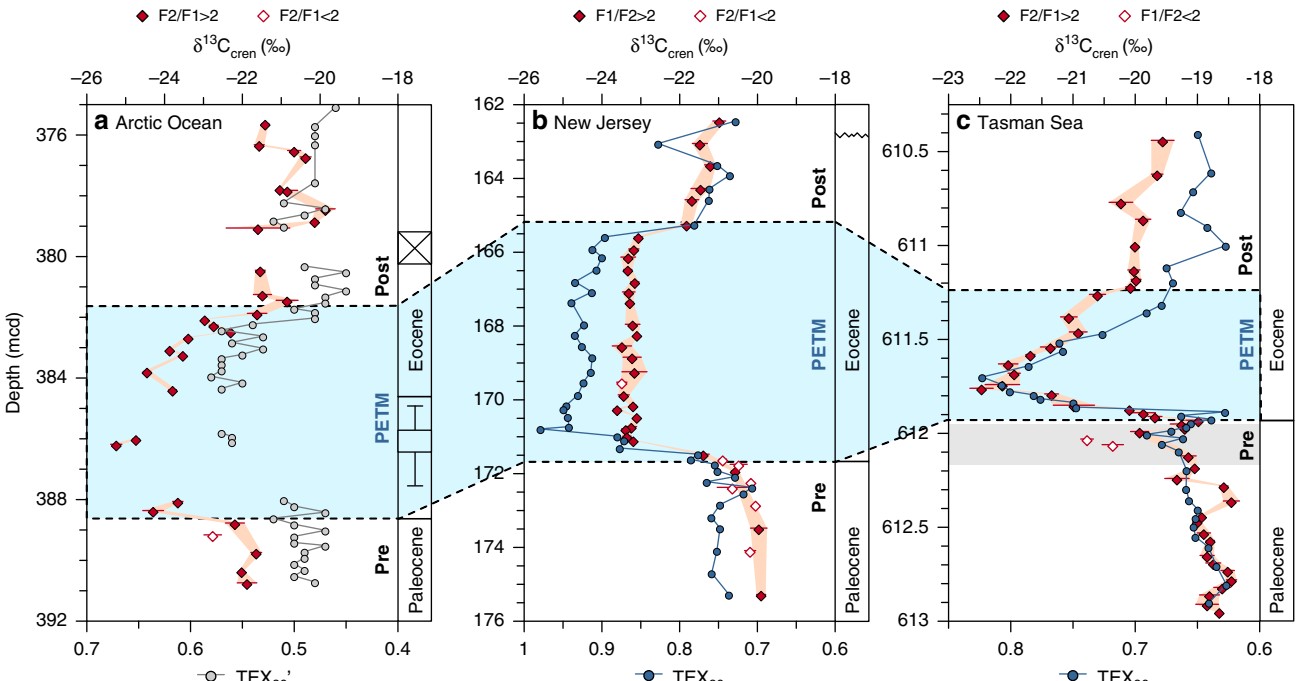

**Fig. 2** Coupled ocean temperature proxy and crenarchaeol stable carbon isotope data across the Paleocene-Eocene Thermal Maximum. TEX$_{86}$ and TEX$_{86}$' index values (plotted on an inverted axis; Arctic Ocean: ref. [43], New Jersey shelf: this study, Tasman Sea: ref. [36]) and carbon isotopic ratios of crenarchaeol (δ$^{13}$C$_{cren}$) during the late Paleocene and early Eocene at three globally distributed sites (**a**, Arctic Ocean; **b**, New Jersey shelf; **c**, Tasman Sea; as in Fig. 1). Error bars for δ$^{13}$C$_{cren}$ represent one standard deviation of replicate analyses ($n = 5$). Error bars of δ$^{13}$C$_{cren}$ are not displayed where they are smaller than the symbol

deep-water Thaumarchaeota did not lead to significant changes in sediment $\delta^{13}C_{cren}$. The consistency of $\delta^{13}C_{cren}$ data through the sapropel events (including pre-event comparison values) imply that strong changes in ecosystem states, such as eutrophication, habitat depth changes, and anoxic/euxinic conditions do not significantly impact $\delta^{13}C_{cren}$ values[53]. This is explained by recognizing that the depth habitat of Thaumarchaeota is linked to oceanographic parameters such as the deep chlorophyll maximum and the depth of maximum remineralization. The thaumarchaeal population should thus maintain a similar relative position along the $\delta^{13}C$ gradient, rather than being fixed to a specific water depth. Although it is possible that warming-induced changes in remineralization rates during the PETM could have impacted $\delta^{13}C_{cren}$ values by changing $\delta^{13}C_{DIC}$ gradients through the thaumarchaeal depth habitat, the extent of these changes likely was minimal given the warm greenhouse background climate of the Paleocene-Eocene[54]. Overall a primary advantage of estimating the $CIE_{DIC}$ from $\delta^{13}C_{cren}$ is the apparent niche stability of marine Thaumarchaeota.

Putative mixotrophy (i.e., autotrophy supplemented by heterotrophic organic carbon assimilation) by marine archaeal consortia could affect values of $\delta^{13}C_{cren}$, assuming autotrophic $\delta^{13}C_{cren}$ is $> -20‰$ and $\delta^{13}C$ of the organic substrate would be $< -20‰$[50,55,56]. However, close examination of earlier studies demonstrates that there is currently no direct evidence for mixotrophy among epipelagic Thaumarchaeota[57]. Previous estimates indicating $17 \pm 11\%$ heterotrophic carbon assimilation were based on suspended particulate matter samples from 700 m water depth[56] and thus were not representative of the major habitat and export depth of Thaumarchaeota, which is centered around ~150 m water depth[58]. Second, previous reports of amino acid utilization[55] solely demonstrated uptake of exogenous amino acids into thaumarchaeal cells but not contribution to biomass carbon. Finally, recent $^{13}C$-label experiments demonstrated that all cultivated Thaumarchaeota are strictly autotrophic[57]. In summary, present knowledge supports the idea that values of $\delta^{13}C_{cren}$ reflect autotrophic growth and reliably record $\delta^{13}C_{DIC}$.

Systematic offsets between $\delta^{13}C_{cren}$ values at our three study sites additionally corroborate an autotrophic origin of crenarchaeol. Preceding the CIE, absolute values of $\delta^{13}C_{cren}$ differ by up to 3‰ between sites, with the most positive $\delta^{13}C_{cren}$ values found in the Tasman Sea and the most negative values observed in the Arctic Ocean, with an intermediate gradient of ~1 to 1.5‰ in $\delta^{13}C_{cren}$ values between the Tasman Sea and New Jersey shelf; the latter gradient is similar in magnitude but opposite in direction to the modern gradient between the North Pacific and North Atlantic[59]. Although lower $\delta^{13}C_{cren}$ values in the northern Atlantic and Arctic compared to the Pacific Ocean support previous arguments for distinct, and potentially reversed, ocean circulation patterns during the Paleocene-Eocene[60–63], similar CIE magnitudes at all sites suggest that the effect of potential deep water circulation reversals on the CIE was minor in comparison to whole-ocean $\delta^{13}C_{DIC}$ changes. Calculating pre-PETM $\delta^{13}C_{DIC}$ from an expected $\varepsilon_{DIC-Cren}$ of $-19.7 \pm 0.1‰$ (Supplementary Fig. 4) yields reasonable $\delta^{13}C_{DIC}$ values for the base of the photic zone of ~$0.8 \pm 0.2‰$ (Tasman Sea), $-0.5 \pm 0.2‰$ (New Jersey shelf), and $-2.2 \pm 0.3‰$ (Arctic Ocean). These $\delta^{13}C_{DIC}$ reconstructions may prove useful as paleoclimate model targets.

The close agreement in CIE estimates at all sites suggests no significant influence of changes in local environmental conditions on CIE expression. Still, comparatively low $\delta^{13}C_{cren}$ values in the Arctic Ocean, when compared to the Tasman Sea and New Jersey shelf, indicate distinct environmental conditions for the Arctic basin. Lower absolute $\delta^{13}C_{cren}$ values may reflect accumulation of $^{13}C$-depleted DIC in the chemocline and anoxic zone of the Arctic Ocean due to reduced mixing under enclosed, euxinic

conditions similar to the modern Black Sea or Cariaco Basin[64]. Further, post-PETM $\delta^{13}C_{cren}$ values in the Arctic Ocean are equal to or higher than pre-PETM $\delta^{13}C_{cren}$ values, while they are lower than pre-PETM values in the Tasman Sea and New Jersey records (Fig. 1). This suggests that during the recovery phase the Arctic Ocean entered a new ecosystem state with a distinct local $\delta^{13}C_{DIC}$ signal. Bulk sediment $\delta^{15}N$ records[65] support the notion that Arctic Ocean ecosystem changes persisted past the peak PETM. Specifically, lower $\delta^{15}N$ values in the PETM recovery phase than during the pre-PETM suggest continued suboxic conditions that were more pronounced than prior to the PETM. However, it remains unconstrained how these ecosystem changes, potentially resulting from variability in ocean stratification, affected local $\delta^{13}C_{DIC}$ values. Because $\delta^{13}C_{DIC}$ dynamics may differ between shelf environments (New Jersey, Tasman Sea), enclosed basins (Arctic Ocean), and deep open ocean environments, additional records from deep ocean environments could help constrain the influence of local effects on $\delta^{13}C_{DIC}$ and $\delta^{13}C_{cren}$. Terrigenous input of crenarchaeol into the proximal Arctic Ocean also could yield depleted $\delta^{13}C_{cren}$ values and a biased $CIE_{cren}$, given that terrigenous crenarchaeol would likely be more depleted in $^{13}C$ than marine-derived crenarchaeol[32]. The BIT record at our deep Arctic Ocean site indicates significant soil input only during the pre-PETM interval (Supplementary Fig. 2). However, it seems unlikely that terrigenous input could significantly impact marine $\delta^{13}C_{cren}$ values as crenarchaeol is a minor or trace component in soils compared to other GDGTs[66]. This view is supported by the consistency of the $CIE_{cren}$ between sites throughout the analyzed time interval regardless of the magnitude of the local BIT value or the absolute $\delta^{13}C_{cren}$ values (Supplementary Fig. 2).

The lack of biases from source heterogeneity and the agreement between all three marine sediment cores yields confidence in our $\delta^{13}C_{cren}$-based $CIE_{DIC}$ estimate of $4.0 \pm 0.4‰$ (Figs. 1, 3). Our $\delta^{13}C_{cren}$ records confirm an earlier $CIE_{cren}$ estimate of $-3.6 \pm 0.3‰$ ($CIE_{DIC} = -4.3 \pm 0.3‰$) using crenarchaeol degradation products from a North Sea record[31], although this previous record is too sparsely resolved to allow rampfit estimates.

Comparison with established proxies suggests that $\delta^{13}C_{cren}$ may circumvent problems that confound other types of $\delta^{13}C$ records. Among the most commonly used proxies, bulk marine, and soil organic matter as well as plant and algal biomarkers tend to overestimate the CIE due to increased isotopic fractionation at higher $CO_2$ levels[67]. Bulk measurements are additionally prone to biases resulting from shifting plant/algal community composition or from varying admixtures of terrigenous and marine organic carbon to the bulk marine TOC pool[19]. These effects are reflected in variable CIE magnitudes of $-3.1‰$ to $-6.7‰$ in $\delta^{13}C_{TOC}$ at our three study sites (Fig. 1). Similarly, CIEs for all compiled records of marine and terrigenous organic matter range from $-1‰$ to $-8‰$ (Fig. 3; ref. [16]). CIEs reconstructed from more specific algal and plant lipid biomarkers likely suffer from similar biases, yielding diverging results of ~$-6.0 \pm 0.2‰$ for plant-derived $C_{27}$ $n$-alkanes but no discernable CIE for putatively algal $C_{17}$ $n$-alkanes at our Arctic Ocean study site[39] (Supplementary Figs. 1, 5). Likewise, CIEs reconstructed from species-specific dinoflagellate cyst $\delta^{13}C$ data from Bass River vary from $-1.8 \pm 0.4‰$ to $-4.0 \pm 0.5‰$ (Supplementary Fig. 5), possibly due to divergent ecological preferences[68]. A plant biomass consensus estimate, which accounts for changes in fractionation induced by precipitation, vegetation, temperature, and altitude yields a continental (or atmospheric) CIE of $-4.6‰$ (ref. [23]). However, this estimate does not account for changes in $pCO_2$-dependent fractionation and thus may overestimate the true CIE.

In contrast to bulk organic matter, bulk marine carbonate and foraminiferal carbonate may underestimate the CIE due to carbonate dissolution in the global deep ocean during the PETM

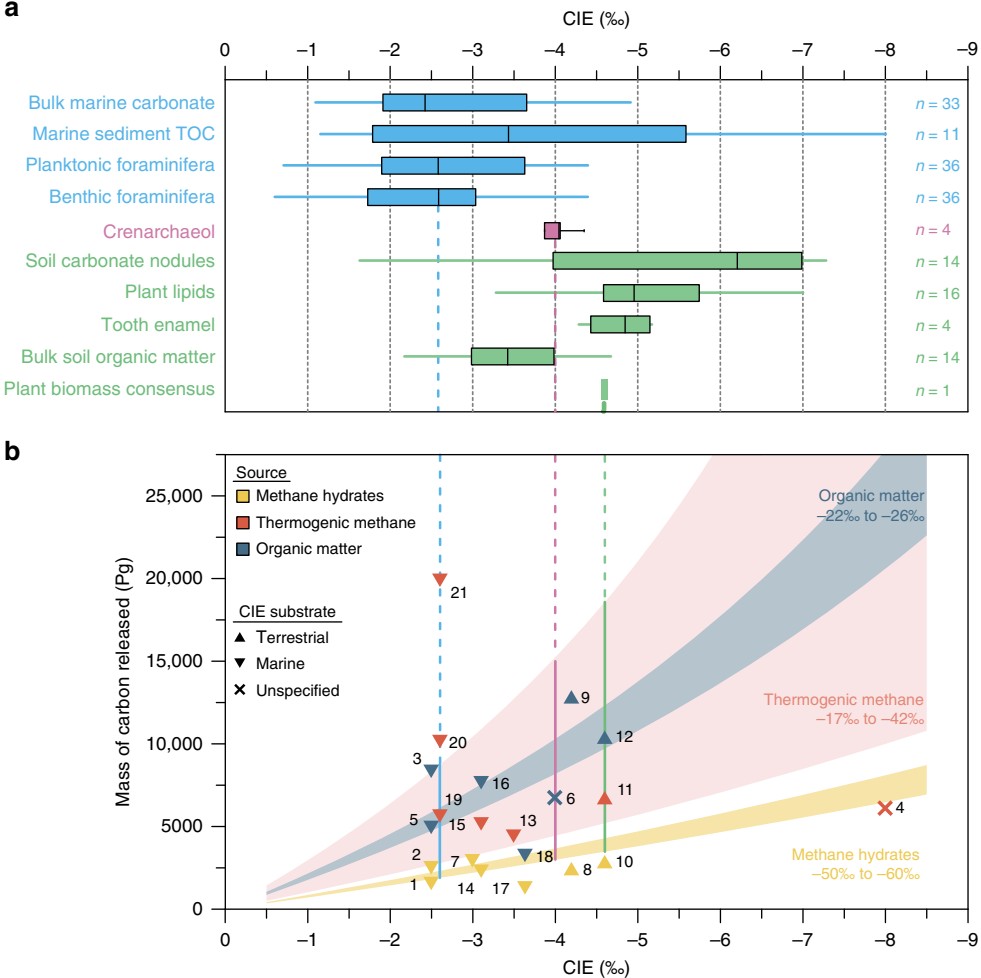

**Fig. 3** Estimates of the carbon isotope excursion and corresponding emission scenarios during the Paleocene-Eocene Thermal Maximum. **a** Carbon isotope excursion (CIE) magnitude for the Paleocene-Eocene Thermal Maximum (PETM) reconstructed from crenarchaeol at our three study sites and one additional site from the North Sea, alongside estimates from other substrates of marine (blue) or terrigenous origin (green) compiled previously[16] (TOC: total organic carbon). Whiskers indicate maximum/minimum values, boxes represent the 25 and 75% quartiles, and the horizontal line identifies the median. **b** Compilation of published PETM emission scenarios with reported CIE value and estimate of total carbon emissions (numbers refer to information listed in Supplementary Data 1; note that studies with multiple scenarios are tabulated and shown in this figure as separate entries), which are superimposed on a parameter space of the $\delta^{13}C$ of the emission carbon source (color shading) defined by the CIE magnitude and the mass of carbon emitted; assuming a fixed mass (40,000 Pg C)[71] and isotopic composition ($\delta^{13}C = -2.5‰$)[78] of the exogenic carbon reservoir. Symbols denote the sample substrate on which the CIE was measured, and color coding indicates the assumed source of carbon in each emission scenario. Vertical bars represent median CIEs recorded in crenarchaeol (this study and ref. [31]), benthic foraminifera[16], and a plant biomass consensus estimate[23]

(ref. [6]; e.g., below ~1500 m water depth in the South Atlantic[25]), and/or post-depositional carbonate recrystallization[26]. In addition, vital effects in foraminifera may result in $\delta^{13}C$ carbonate-seawater disequilibria[27,69,70]. These confounding effects are reflected in a large spread of foraminifera-based CIE estimates between −0.6‰ and −5.1‰ for benthic species (median −2.6‰)[16] and −0.7‰ to −4.4‰ for planktonic species (median −2.6‰)[16]. At the New Jersey shelf, the only site studied here with well-preserved (glassy) foraminifera, benthic foraminifera indicate a CIE of −3.5 ± 0.3‰ (Ancora[37]) and −3.8 ± 0.4‰ (Bass River[46]; Supplementary Fig. 1), while the CIE derived from planktonic foraminifera (Bass River) is somewhat more variable (−3.6 to −4.3‰)[46]. Along with other CIE estimates from well-preserved foraminifera from the South Atlantic (−3.4 ± 0.2‰, paleo-water depth of ~1500 m; Supplementary Fig. 1)[25] and the North Atlantic (−3.3 ± 0.2‰; Supplementary Fig. 1)[37], these estimates are consistent with our $\delta^{13}C_{cren}$-based $CIE_{DIC}$ estimate, emphasizing that the global median CIE of foraminiferal records of −2.6‰ (ref. [16]) likely is

biased by generally poor carbonate preservation. Additional biases from vital effects[69,70] may have amplified (e.g., decrease in $\delta^{13}C$ of the food source) or dampened (e.g., decrease in carbonate ion concentration) the CIE in foraminifera. However, it is currently not possible to quantify the net bias resulting from these effects.

The observation of similar CIE magnitudes in both benthic foraminifera and planktonic archaea, which broadly represent the bathypelagic and epipelagic DIC pools, respectively, suggests complete propagation of the CIE signal from the surface into the deep ocean within the temporal resolution of our records (> 3–4 ka). Equilibration of the surface and deep ocean carbon pools within < 4 ka is supported by model simulations across a wide range of carbon emission rates, magnitudes, and isotopic compositions[18]. Accordingly, paleorecords with higher temporal resolution could yield larger initial CIEs that were not equilibrated across the entire exogenic carbon reservoir[18]. Indeed, very negative $\delta^{13}C$ values at the PETM onset were observed in single

foraminifera[26] and putatively annually-resolved bulk carbonate records[28]. Taken together, the $\delta^{13}C_{cren}$ records narrow down the fully-mixed marine CIE to $-4.0 \pm 0.4‰$. As most exogenic carbon is stored as DIC in the ocean and is mixed with the other sub-reservoirs on millennial timescales, we argue that this estimate may approximate the average total CIE for the exogenic carbon reservoir.

The magnitude of the CIE reconstructed from $\delta^{13}C_{cren}$ can be used to refine plausible carbon emission scenarios for the PETM. To this end, we compiled previous estimates of CIE magnitude as well as the mass and isotopic composition of emitted carbon (Fig. 3b, Supplementary Data 1). Based on assumed or measured CIEs of $-2.5$ to $-5.5‰$, these previous estimates of carbon emissions vary widely from 1100 Pg to 20,000 Pg C. Our CIE estimate of $-4.0 \pm 0.4‰$ is significantly larger than predicted from most marine records and is therefore also larger than CIEs used in most mass balance calculations and carbon cycle models[8,13,16,17]. Mass balance calculations using our CIE estimate (Fig. 3b) constrain emissions to 4400–15,200 Pg C if the carbon source is thermogenic methane and $CO_2$ ($\delta^{13}C = -17‰$ to $-42‰$)[17,20], to 3000–3700 Pg C if the source is only methane (e.g., hydrates; $\delta^{13}C = -50‰$ to $-60‰$)[8,13,22,71], and to 8200–10,300 Pg C if the source is organic matter ($\delta^{13}C = -22‰$ to $-26‰$)[9,14,21]. These ranges of $\delta^{13}C_{cren}$-based emissions estimates are not only narrower than previous mass balance estimates but also more precise, with an uncertainty of ~10% compared to the much larger uncertainties resulting from the wide scatter of other substrate-specific CIEs[16] (Fig. 3a).

Although a mass-balance approach assumes a single emission pulse, our $\delta^{13}C_{cren}$-based $CIE_{DIC}$ estimate also allows for re-evaluation of previous model-derived estimates that incorporate more complex features such as sustained emissions and carbon burial. It is notable that many of the more complex modeling approaches to date have been based on smaller magnitudes for the CIE of $-2$ to $-3‰$[13,17]. For instance, coupled carbonate dissolution data-model simulations suggest emission of <3000 Pg C (ref. [13]), with a CIE of $-3‰$ in foraminifera used to independently constrain the $\delta^{13}C$ of the carbon source to $-50‰$. The $-4.0 \pm 0.4‰$ CIE in crenarchaeol instead implies that the carbon isotopic composition of the emitted carbon would be required to be more negative (< $-60‰$) for this scenario. Similarly, ocean pH data-model comparisons[17] suggesting larger emissions (5700–20,000 Pg C) used a CIE of $-2.6‰$ derived from planktonic foraminifera to constrain the $\delta^{13}C$ of the carbon source to $-11‰$ (ref. [17]), implying emission of mostly mantle-derived $CO_2$. Our larger $\delta^{13}C_{cren}$-based CIE again would imply a more $^{13}C$-depleted carbon source; specifically, it would scale the mean emission $\delta^{13}C$ value to $-17‰$ (using a CIE value of $-4‰$ but otherwise following the approach of Gutjahr et al.[17]). A revised emission $\delta^{13}C$ value of $-17‰$ suggests larger contributions from organic matter, such as from thermogenic methane generated by North Atlantic sill emplacement[17,20,72], sedimentary organic matter oxidation[9,21,73], or methane hydrate dissociation[8]. In summary, the tighter constraints on the CIE from $\delta^{13}C_{cren}$ suggest that previous estimates based on smaller CIEs may have pointed to carbon source $\delta^{13}C$ values that are too enriched in $^{13}C$ and/or underestimated the magnitude of carbon emissions. Carbon emissions must have been larger than 3000 Pg C for a release from methane hydrates and larger than 4400 Pg C for any scenario involving contributions from sedimentary organic matter oxidation and/or geothermal heating. Building on our refinement of the CIE magnitude and uncertainty, emerging highly resolved records of $pCO_2$ or carbonate system parameters (e.g., [DIC]) will allow closer evaluation of the rates and mechanisms of PETM carbon emissions.

Analysis of archaeal lipids further provides coupled temperature data via the $TEX_{86}$-paleothermometer, which can be used to evaluate the role of pre-CIE warming as a trigger for carbon emissions. This approach avoids biases from diagenetic carbonate overgrowth and dissolution affecting foraminiferal $\delta^{13}C$ values[26] and stratigraphic ambiguities that arise when separate substrates are used for the CIE and warming. Although incompleteness of the PETM onset in the Arctic Ocean record, and potentially the Tasman core, as well as a lack of well-constrained age models limit the determination of lead-lag relationships, the parallel changes in $\delta^{13}C_{cren}$ and $TEX_{86}$ (Fig. 2) suggest that any warming preceding the CIE must have occurred no more than 3–4 ka prior, consistent with previous estimates from the New Jersey shelf[38,74], the equatorial Atlantic[74], and the Southern Ocean[74]. The coupled $\delta^{13}C_{cren}$-$TEX_{86}$ approach described here could be instrumental for determining high-resolution lead-lag relationships of temperature and CIE during the PETM and other climate events.

The coupled $\delta^{13}C_{cren}$-$TEX_{86}$ records further yield important insights into the functioning of the carbon cycle immediately prior to the PETM. Our highest resolution record from the Tasman Sea captures a decrease of ~ $-1‰$ in $\delta^{13}C_{cren}$ and a coeval increase in $TEX_{86}$ (equivalent to 1–2 °C warming, depending on calibration; Supplementary Fig. 6) preceding the PETM (Fig. 2). This event may be correlated to a 3‰ decrease in $\delta^{13}C$ of pedogenic carbonates, the pre-onset event (POE), recorded in a terrestrial section from the Bighorn Basin (Wyoming, USA)[12]. If these events were indeed coeval, the POE must have been a global event. The POE was an ephemeral event that lasted < 2 ka before recovery to pre-POE conditions[12], as recorded in the decrease of both $TEX_{86}$ and $\delta^{13}C_{cren}$ values in the Tasman Sea immediately prior to the PETM onset (Fig. 2). Similarly, $\delta^{13}C$ values decrease to pre-POE values before the PETM onset in the Bighorn Basin[12]. In contrast to the PETM, which must have been sustained by continuous emissions to explain its protracted recovery[75], the POE could have originated from a single carbon release, such as from gas hydrates. The rapid recovery suggests that critical thresholds for positive feedback mechanisms were not reached.

Due to their contrasting behavior, high-resolution $\delta^{13}C_{cren}$ and $TEX_{86}$ studies of the PETM and other Paleogene hyperthermals, including the POE, may yield constraints on the relative timing of warming, carbon emissions, and recovery. The diagenetic stability and potential for high-resolution records of archaeal lipids suggest that $\delta^{13}C_{cren}$ is a promising proxy, particularly for reconstructing transient CIEs that exhibit poor carbonate preservation, such as during Mesozoic Oceanic Anoxic Events and early Cenozoic hyperthermals. Further ground-truthing using a wider range of thaumarchaeal cultures and globally representative environmental datasets will help further constrain uncertainties of the $\delta^{13}C_{cren}$ proxy and will strengthen its application to the paleoenvironment.

## Methods

**Lipid extraction.** Sediment samples (10–20 cm³) were freeze dried, homogenized, and subsequently extracted in teflon vessels in three steps (50:50 dichloromethane: methanol, 90:10 dichloromethane:methanol, 100% dichloromethane) using a MARS5 microwave-assisted extraction system (CEM Corporation, Matthews, NC, USA). The microwave program consisted of 30 min heating to 100 °C and 20 min hold time, after which the solvent was decanted and fresh solvent was added. The extracts from the successive extraction steps were combined into a total lipid extract (TLE), dried under $N_2$, and stored at $-20$ °C until measurement. The TLEs were split into two aliquots for (i) purification and carbon stable isotopic analysis of crenarchaeol and (ii) analysis of lipid abundances for determination of $TEX_{86}$ and BIT.

**Purification and carbon isotopic analysis of crenarchaeol.** TLE aliquots were separated over pre-combusted $SiO_2$ (130–270 mesh) to yield a GDGT-containing fraction[32]. The GDGT-containing fraction was further purified by orthogonal

semi-preparative high performance liquid chromatography (HPLC) on an Agilent 1200 series HPLC equipped with a fraction collector, following previously described protocols[32,53].

The carbon stable isotopic composition of crenarchaeol was analyzed using a spooling wire microcombustion–isotope ratio mass spectrometer (SWiM–IRMS)[32]. The fractions were dissolved in ethyl acetate and 30–100 ng were manually injected onto the wire (3 replicate injections for F1, 5–6 replicates for F2). The raw data were corrected for blanks and absolute offsets using dilution series of the $C_{46}$-GTGT standard. The $1\sigma$ precision averaged over all samples was 0.15‰. Purity of the crenarchaeol fraction was assessed using the ratio of SWiM–IRMS F2/F1 peak areas (Supplementary Data 2). Analysis of F2/F1 versus $\delta^{13}C_{cren}$ shows that F2/F1 ratios smaller than 2 yield negatively biased $\delta^{13}C_{cren}$ values, due to generally more negative $\delta^{13}C$ values of the background organic matter relative to crenarchaeol. Therefore, only data with F2/F1 ratios >2 were used for calculating the CIE magnitude. All isotopic ratios are reported relative to the Vienna Pee Dee Belemnite scale.

**Analysis of lipid abundances**. Relative abundances of GDGTs (Supplementary Data 3) from site ODP 174AX were determined from aliquots of the same TLE used for crenarchaeol isotopic analyses, using the instruments and conditions described in ref. [53].

$TEX_{86}$ and BIT values were calculated from relative abundances of GDGTs for the Ancora record. Published $TEX_{86}$ and BIT data from adjacent samples were used for the Arctic Ocean[43] and Tasman Sea records[36]. $TEX_{86}$ values were corrected for differences in peak separation between the traditional single-column HPLC-MS method and the newer protocol[76] employed here by using a cross-calibration of 27 Arabian Sea and Atlantic Ocean sediments. Due to the ambiguity associated with choosing appropriate deep-time $TEX_{86}$ sea surface temperature calibrations, all data are presented as $TEX_{86}$ or $TEX_{86}'$ ratios rather than putative sea surface temperatures, with the understanding that higher ratios correspond to relatively warmer temperatures.

**Carbon isotopic analysis of bulk sediment**. Content and carbon isotopic composition of TOC at sites ODP 174AX and IODP 302 (Supplementary Data 4) were analyzed on acidified bulk sediment aliquots using a ThermoScientific Flash EA Delta V Plus IRMS. Carbon isotopic compositions were peak-size-corrected and offset-corrected using laboratory and authentic reference standards (USGS40, USGS41a). The standard deviation of $\delta^{13}C_{TOC}$ as determined by replicate analysis of reference standards was 0.04‰.

**Calculation of carbon isotope fractionation and carbon isotope excursion**. $\varepsilon_{DIC-Cren}$ was calculated for pre-PETM and peak-PETM boundary conditions (Supplementary Fig. 4) using the flux balance model by Pearson et al.[41]. Boundary conditions[13,17] were set as: 25 °C seawater temperature, salinity 35, [DIC] 2000–3000 μmol kg$^{-1}$, pH 7.75–7.45. $\varepsilon_{DIC-Cren}$ depends on assumed $pCO_2$ change across the PETM, following results from Gutjahr et al.[17], $\varepsilon_{DIC-Cren}$ changed by 0.9‰, from of 19.7 ± 0.1‰ for the pre-PETM to 18.8 ± 0.05‰ for the peak-PETM. However, using the smaller $pCO_2$ change suggested by Zeebe et al.[13], $\varepsilon_{DIC-cren}$ changed by only 0.6‰. Uncertainty in $pCO_2$ change across the PETM thus leads to an uncertainty in $\varepsilon_{DIC-cren}$ of ± 0.15‰ from an average of 0.75‰. $CIE_{cren}$ was calculated from pre-PETM and peak-PETM $\delta^{13}C_{cren}$ data using the rampfit software[45]. To derive $CIE_{DIC}$, pre-PETM and peak-PETM $\varepsilon_{DIC-cren}$ were applied to $CIE_{cren}$ to derive $CIE_{DIC}$ and uncertainty was determined by propagating the measurement uncertainty, the rampfit estimated error of $CIE_{cren}$, and the uncertainty in $\varepsilon_{DIC-cren}$.

### Data availability
All data underlying main text and supplementary figures are available as a Source Data file and have been deposited in the Pangaea repository (https://doi.org/10.1594/PANGAEA.900498) and the Harvard Dataverse (https://doi.org/10.7910/DVN/OJJGYN).

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

## Acknowledgements

This research used samples provided by the Ocean Drilling Program (ODP) and the International Ocean Discovery Program (IODP) and was supported by the National Science Foundation (Grants 1702262 and 1843285, to A.P.), the Gordon and Betty Moore Foundation (A.P.), the Deutscher Akademischer Austauschdienst (RISE World-wide Fellowship #US-ES-2997 to K.D.D.) and the Deutsche Forschungsgemeinschaft (KU2842/1-1, to S.K.; and GO2294/2-1, to J.G.). Laura Kattein is thanked for assistance with sample preparation.

## Author contributions

F.J.E., A.P., and S.K. designed the study. F.J.E., K.D.D., A.P., S.K., S.J.H., and J.G. performed the analyses. F.J.E., A.P., and J.G. wrote the paper with contributions from all authors.

## Competing interests

The authors declare no competing interests.
