## [Peer Review File · Nature Communications]

Reviewers' comments:

Reviewer #1 (Remarks to the Author):

Dear Dr. Collins,

Please find below my remarks to the paper by Elling and colleagues. The authors are to be congratulated on their analytical skills. It is quite amazing to see stable carbon isotope ratios of crenarchaeol preserved in Paleocene and Eocene sediments. These values are of interest to the field of marine biogeochemistry, both for the modern and paleocommunities.

I am slightly less enthusiastic about the interpretation of the data and the implications the authors extract from these. First, the authors imply that the fractionation between DIC and $\delta^{13}\text{C}$ -cren is constant across the PETM. This assumption is strange and not substantiated by published literature. It is to be expected that other factors, such as temperature and pH, affect fractionation (Könneke et al., 2012; ref 32 in the present MS) and both of these factors change tremendously across the PETM. Secondly, the authors assume that ^{13}C fractionation is identical for all ammonia-oxidizing archaea; this may be true but it is currently untested. Thirdly, the authors assume that bicarbonate is the only source of carbon in Thaumarchaeota, while it has been suggested that there might be a contribution of heterotrophy that would significantly affect $\delta^{13}\text{C}$, as admitted in the manuscript. Fourthly, the authors assume that the change in $\delta^{13}\text{C}$ of DIC that is locally acquired by the archaea at the study sites reflects the change in $\delta^{13}\text{C}$ of the global exogenic carbon pool. This is problematic because at all the study sites changes in ecology, hydrological regime and production should have led to changes in vertical water column structure, vertical $\delta^{13}\text{C}$ gradients of DIC and the dominant depth of crenarchaeol production (some of the relevant papers are cited by the authors). Moreover, the $\delta^{13}\text{C}$ of DIC in such coastal sites are unlikely to reflect the change in the global exogenic carbon pool because it is unlikely that the spatial distribution of stable carbon isotopes remained constant globally, given the nature and severity of global climate, ecological and biogeochemical change. Only the deep ocean might sample something close to the global exogenic CIE but that record is compromised by carbonate dissolution (the one seemingly complete record has been suggested to be biased from circulation change; McCarren et al., 2008; ref 26). In other words, the general philosophy of this study suffers from an overly reductionist starting point.

Collectively, I disagree with the conclusion that the 3.5 ‰ CIE found at three study sites must reflect the magnitude of the CIE in the global exogenic carbon pool.

Below I include my comments chronologically with the manuscript.

27. here 'early paleogene' seems a stretch as it regards the interval between 66 and 34 million years ago. The hyperthermals are within a much more narrow age band.

30. consider if methane in clathrates can properly be referred to as fossil organic carbon

33/34. Here a unidirectional causal relation is inferred: emissions cause warming. But reference 5 suggested this was the other way around.

44-61. This problem was systematically approached by Sluijs and Dickens (your ref 43). From their abstract: 'Individual carbon isotope records differ in shape and magnitude from variations in the global exogenic carbon cycle through changes in (1) the relative abundance of mixed components with different $\delta^{13}\text{C}$ within a measured substrate, (2) isotope fractionation through physiological change, and (3) the isotope composition of the carbon source. All three factors likely influence many early Paleogene $\delta^{13}\text{C}$ records, especially across the PETM and other hyperthermal events.' Particularly because this paper claims to have found the true value of the CIE, these three problems need to be excluded.

69. The cited paper (ref 32) reports mean fractionation values of about 19.5 to 19.8. This is an important number but also, The 19.5 – 19.8 values are mean values without uncertainty so here you want to include an uncertainty analysis based on the available data.

70-71. Thaumarchaeota predominantly reside at the base of the photic and upper sub-photoc zones (ref 37,38).

Much evidence, from both modeling and field data suggest that water column structure and productivity changed significantly during the hyperthermals (I presume Carmichael et al., 2017 summarize this evidence but I have not checked). This could affect the result in two ways. 1) it could have affected the DIC-d13C gradient from surface to deep (Sluijs and Dickens issue 3), and 2) it might have affected the archaea thriving in the subsurface; they may have migrated to greater or lesser depth (Sluijs and Dickens issue 3), there could have been a change in communities (Sluijs and Dickens issue 1) or they change their fractionation (Sluijs and Dickens issue 2). If you can address all of these potential errors, you arrive at the local expression of the CIE of DIC; then the step to the global exogenic one needs to be made.

81-83. So if this is all true the uncertainty is now between 2,200 and 11,000 Pg. Is that a big improvement?

93-95. Awkward phrasing; suggests that the Lomonosov Ridge was a restricted basin.

96-98. This is different than most previous approaches, where the difference between average so-called 'body' CIE values and pre-CIE values is taken. Why choose the minimum value here rather than a mean of multiple analyses with a proper statistical measure of uncertainty? It is indicated that the reported uncertainty includes analytical uncertainty but not how this stratigraphic issue is managed.

101-102. Is this just the average of the three numbers with the propagated uncertainties?

106-108. It is relevant here to compare the Arctic result to previous work, notably that of the short chain alkanes of Pagani et al., (2006, Nature; currently not cited) and that of the synthetic marine organic carbon record from ref 43?

108-109. At the New Jersey shelf, the only site studied here with sufficiently preserved carbonates, benthic foraminifera indicate a CIE of $-3.4 \pm 0.14\text{‰}$ (ref. 44).

It is generally accepted that seawater carbonate chemistry affects carbon isotope ratios in foraminifera (e.g., Spero et al., 1997 and follow-up work). This has been explored for the PETM by ref 57, at least for planktic foraminifera, and found to dampen the CIE. If so, why is the local CIE in archaeal lipids not larger?

Figure 1.

- here the d13C-cren results are compared to d13C-TOC. Why? For both the Arctic Ocean and New Jersey sites, d13C records of more specific marine substrates have been generated that allow for a better comparison.

- I assume that the d13C-cren data were generated on the same extracts used for TEX86 analyses by the original papers but this is mentioned nowhere.

128. BIT is considered a tracer of soil organic matter as far as I know.

128-130. Strange sentence

128-144. This was all extensively discussed in previous work; why repeat here (in lines 128-139 even without referring to that work)? It is in the results section, which makes it extra weird because it regards data generated in previous work (I think except for this New Jersey record).

157. If it is 100% certain that Thaumarchaeota fix ONLY bicarbonate, that should be specified here. If it is uncertain if also other carbon sources may be incorporated (CO₂ or some degree of heterotrophy), that should be mentioned here too.

161. Carbon isotopic fractionation in thaumarcheota is understudied and the constraints may be insufficient to make the claims that are presented in this paper. Line 159-160, for example, indicates that the authors don't expect differences between clades but with the present data, there is no way to evaluate that claim. In addition, I am missing a critical evaluation of potential other factors influencing ^{13}C fractionation. It is important to note that ref 32 studied if the biphytanes are a decent proxy for bulk biomass in the species *Nitrosopumilus maritimus*, not if the fractionation between DIC and biomass is constant. Ref 32 explicitly expects changes in fractionation as a function of temperature and pH. These factors are currently ignored while both factors vary between study sites and change dramatically during the PETM.

164. Isn't there earlier work that already showed this than ref 47?

165-167. I disagree with this statement as it assumes that things don't change at the PETM. They do in a relevant way. For both New Jersey and the Arctic, big changes in the hydrological regime have been recorded and that should affect water column structure and productivity and that has been described in several papers for both regions. It is therefore unlikely that the $\delta^{13}\text{C}$ gradient of DIC and the depth of origin of the isoprenoid GDGTs remained the same. This is important because it directly affects the main conclusion.

168-170. This is a major issue and I disagree with the way this is disregarded as a potential problem. All micropaleontological studies indicate a massive change in ecosystems during the PETM. Here it is stated that a significant portion of the GDGT carbon atoms may derive from some food source. However, during the PETM the $\delta^{13}\text{C}$ of that food source will not change with the same magnitude as the $\delta^{13}\text{C}$ of DIC (that is the implicit assumption, not the $\delta^{13}\text{C}$ values that are given here). This fact is actually the motivation of this study. Even if the uncertainty is 1‰ or less, this probably implies that this uncertainty must be propagated in the magnitude of the CIE that is reconstructed. Why is it not?

170-173. Indeed, so if this is unknown, how can we really tell that you're correct? Seems very opportunistic.

174. Strange sentence. What does 'its' refer to?

183. Ref 35 reports higher values; important here to include the full uncertainty as mentioned above.

186. What is a 'basinal environment'?

187. Are there indications that euxinic conditions prevailed in the Arctic prior to the PETM?

188-189. Why would this lead to an overestimate of the CIE? As BIT drops during the PETM shouldn't it lead to an underestimate?

191-200. Consistency between the sites cannot be used as an argument. All potential confounding factors should be dealt with at the individual sites. Then it should be evaluated if consistency between 3 sites gives you the global exogenic CIE or not.

204-206. If so, than this affects the potential heterotrophic contribution to the GDGTs.

209-212. I don't understand this; both factors have been shown to lead to larger CIEs in organic matter (see, e.g., plant community shift study of Schouten et al., 2007 EPSL)

Figure 3.

- the bulk marine organic matter data; isn't that rather TOC from marine sediments (so really a mixture of marine and terrestrial organic carbon)?

237-243. Not if a pH effect is added to the foram records

241. The benthics from ref 26 are heavily dissolved and recrystallized. Far from 'glassy'.

246-247. Why is this a strong point?

248-249. Why is the CIE not compared with planktic or thermocline-dwelling foraminifera at New Jersey? They may be closer to the position of the archaea than the benthics. Then you don't have to assume that the $\delta^{13}\text{C}$ depth gradient remained identical during the PETM (which it probably didn't).

260 Most of the recent modeling regarding the mass of injected carbon during the PETM has focused on the total mass of carbonate dissolution (Panchuk, Zeebe, Ridgwell, Cui etc). This has constrained the mass a little better than the wide range suggested here.

261. Is the number 34,000 correct? Seems small (smaller than at present)

266. It depends on a little more than that if methane hydrates were involved (Dickens, 2011 CP) given the time scale of steady state disruption.

271 and further. This exercise is identical to that presented in multiple papers.

277. Please do read the Dickens papers carefully (1995 Paleocyanography and 2011 CP) for a more nuanced statement on this. The methane hydrate hypothesis needs warming to release the methane as a positive feedback. It was never an explanation for the warming until Pagani et al (2006) thought it was and subsequently excluded it.

302-304. I don't think any of the papers that have suggested pre-CIE warming have suggested this for your study sites. Your Arctic record has no recovery for the onset of the CIE; your southern ocean record has very low time resolution; your new jersey record may well be incomplete at the onset. So how can you really study this question?

305. High sampling resolution but the thickness of the CIE there, following your figure 1, is less than a meter and the onset perhaps condensed due to sea level rise (ref. 40). So that is why a large community has focused on several New Jersey shelf sites, where sedimentation rates are more than 10 times higher.

308-310 I fundamentally agree with this point. To have full added value, though, sites where lead/lag relations have been identified should be tested.

313-314. The Arctic Ocean site has no recovery across the onset; the top of Core 32X represents caved sediments (Pagani et al., 2006). Events similar to Bowen et al's POEs actually have been found in other New Jersey cores.

329. but the present CIE has not been mixed with the entire global exogenic carbon pool, notably the deep ocean! And based on what is the termination of the present perturbation more similar to POE's than to the PETM? If the POEs are truly global carbon cycle perturbation, we really do not understand how they ended so quickly. See papers by Archer on the long-term legacy of antropogenic carbon.

336-337. I disagree. The present records suggest instantaneous recovery from POE's while, e.g., ETM2 recovery was in the order of 40 kyr (e.g., Stap et al., 2009 Paleocyanography).

339-340. It is probably important to read the literature on PETM deposition along the New Jersey shelf in a bit more detail. It is well possible that the studied core is incomplete at the onset. Other cores show pre-CIE warming.

399-400. Do TEX86 values correspond well to the previously studied PETM sections from this region?

Reviewer #2 (Remarks to the Author):

I have read through the manuscript (MS) by Elling and colleagues several times. The MS tangentially addresses several issues in the geosciences, including one of the top current and cool problems facing our community, namely how can massive amounts of carbon enter the ocean and atmosphere really fast in the past, such as across the Paleocene Eocene Thermal Maximum (PETM). This was a geologically brief time of major warming and environmental change, and serves as perhaps our best past analog for anthropogenic changes.

The authors measure the $\delta^{13}\text{C}$ of crenarchaeol at three sites across the PETM. This is novel and sort of cool. They show that the CIE excursion in these records is about -3.5 per mil.

Per background, and as noted at the start of their MS, a prominent negative CIE manifests in all carbon-bearing phases across the PETM. For 23 years, this has suggested as representing a massive input of ^{13}C -depleted carbon into the exogenic carbon cycle (ocean, atmosphere and biomass) (Dickens et al., 1995). The source of this carbon has remained somewhat controversial, in part because the magnitude of the CIE varies between locations and substrates analyzed.

Several papers have argued that the CIE expressed in certain records represent global changes in the carbon isotopic composition of the entire exogenic carbon cycle. This has never made sense to me, but remains circa 2018 a "debatable" view. We thus have papers invoking mechanisms to explain a <3.0 per mil $\delta^{13}\text{C}$ excursion to >5.0 per mil $\delta^{13}\text{C}$ excursion

I have mixed views of the MS. On the one hand, the data is novel and interesting; on the other hand I personally learned nothing new in the big picture. But this is perhaps because I have worked on the problem for over two decades and it always has been clear to me that the CIE must be around 3.0 per mil, and deviations from this value in various substrates represent one of several processes (Sluijs and Dickens, 2012). There are also some awkward issues and numerous minor items. I note these on an annotated document and below.

I think the MS needs several significant modifications before publication, but with amendments could make a nice contribution.

Sincerely,

Gerald (Jerry) Dickens

Issues elaborated on from annotated manuscript

(A) The manuscript does not fully capture and explain key issues. The biases and discrepancies lie in individual CIE measurements and how they relate to changes in the exogenic carbon cycle. This issue arises in multiple places and has to be amended.

(B) There is a nominally 19 per mil fractionation between DIC and crenarchaeol (Figure 1 and Line 69). However, it's not crystal clear in the text (Lines 69+, and 155+) how and why this fractionation occurs. Note here that DIC includes both HCO_3^- and CO_3^{2-} and the pH across the PETM changed (as well as likely the relative proportion of HCO_3^- and CO_3^{2-}). So, why would the fractionation remain constant? This is confounded by the issue noted on Line 169 that there be mixing. Albeit < 1 per mil, this is huge in the context of the MS goals. Even assuming that the fractionation remained constant, why would the $\delta^{13}\text{C}$ of the lower photic maintain a constant offset to the exogenic carbon cycle?

This is sort of addressed from an observational point later (Line 194+).

(C) I do not understand Figure 2 in the details. The relationship between theoretical carbon inputs

and $\delta^{13}\text{C}$ has been discussed many times since 1995, although I think Pagani et al. (2006) were the first to put into such a conceptual diagram. Obviously, the relationship depends on the mass and $\delta^{13}\text{C}$ of exogenic carbon cycle. The pCO_2 relationship (right axis) is not so simple and depends on multiple factors. More crucially, I do not understand the points, as many of these references discuss multiple possibilities.

(D) It seems that the authors also measured $\delta^{13}\text{C}$ of TOC, but this is not really emphasized in the Introduction and results. Indeed, until I read the methods and re-read the caption to Figure, I assumed that these values came from other works. Here it important to note that $\delta^{13}\text{C}$ of TOC and other phases (carbonate) have been measured at all three sites. But a proper comparison is missing.

(E) There is a lot of "arm-waving" in the last fourth of the paper (Line 271+) on topics that are not solved in the current MS but have been discussed extensively in the literature. I would much prefer if the authors expanded upon the above issues and toned down the speculation and rehashing at the end. I did not learn anything new here.

Revisions and full point-by-point response to reviewers' comments on manuscript NCOMMS-18-28519-T
"Archaeal lipid biomarker constraints on the Paleocene-Eocene carbon isotope excursion" by F.J. Elling, J. Gottschalk, K.D. Doeana, S. Kusch, S.J. Hurley, A. Pearson

We thank the two reviewers for the thorough evaluation of our manuscript. Their comments allow us to clarify the findings of our study and highlight the value of our newly introduced crenarchaeol $\delta^{13}\text{C}$ proxy for dissolved inorganic carbon (DIC) $\delta^{13}\text{C}$ reconstructions. We revised the manuscript as described point-by-point below. Both reviewers highlight the novelty of our approach, its potential for paleoclimate reconstructions and its value for a broad range of readership. However, their criticisms converge at two major aspects, namely that of potential differences between the local and global carbon isotopic excursion (CIE), and of potential variations in the fractionation factor (ϵ) between $\delta^{13}\text{C}$ of crenarchaeol and $\delta^{13}\text{C}$ of DIC in the ocean. We are confident that we have addressed these concerns adequately in the revision.

Reviewers' comments:

Reviewer #1

Dear Dr. Collins,

Please find below my remarks to the paper by Elling and colleagues. The authors are to be congratulated on their analytical skills. It is quite amazing to see stable carbon isotope ratios of crenarchaeol preserved in Paleocene and Eocene sediments. These values are of interest to the field of marine biogeochemistry, both for the modern and paleocommunities.

I am slightly less enthusiastic about the interpretation of the data and the implications the authors extract from these.

i) The following points by the reviewer all relate to understanding controls on the fractionation factor, ϵ , between DIC and archaeal lipids.

First, the authors imply that the fractionation between DIC and $\delta^{13}\text{C}$ -cren is constant across the PETM. This assumption is strange and not substantiated by published literature. It is to be expected that other factors, such as temperature and pH, affect fractionation (Könneke et al., 2012; ref 32 in the present MS) and both of these factors change tremendously across the PETM. Secondly, the authors assume that ^{13}C fractionation is identical for all ammonia-oxidizing archaea; this may be true but it is currently untested. Thirdly, the authors assume that bicarbonate is the only source of carbon in Thaumarchaeota, while it has been suggested that there might be a contribution of heterotrophy that would significantly affect $\delta^{13}\text{C}$, as admitted in the manuscript.

ii) The following points by the reviewer all relate to oceanographic expression of the CIE and the potential for regional heterogeneity.

Fourthly, the authors assume that the change in $\delta^{13}\text{C}$ of DIC that is locally acquired by the archaea at the study sites reflects the change in $\delta^{13}\text{C}$ of the global exogenic carbon pool. This is problematic because at all the study sites changes in ecology, hydrological regime and production should have led to changes in vertical water column structure, vertical $\delta^{13}\text{C}$ gradients of DIC and the dominant depth of crenarchaeol production (some of the relevant papers are cited by the authors). Moreover, the $\delta^{13}\text{C}$ of DIC in such coastal sites are unlikely to reflect the change in the global exogenic carbon pool because it

is unlikely that the spatial distribution of stable carbon isotopes remained constant globally, given the nature and severity of global climate, ecological and biogeochemical change. Only the deep ocean might sample something close to the global exogenic CIE but that record is compromised by carbonate dissolution (the one seemingly complete record has been suggested to be biased from circulation change; McCarren et al., 2008; ref 26). In other words, the general philosophy of this study suffers from an overly reductionist starting point.

Collectively, I disagree with the conclusion that the 3.5 ‰ CIE found at three study sites must reflect the magnitude of the CIE in the global exogenic carbon pool.

Below I include my comments chronologically with the manuscript.

We thank Reviewer #1 for the detailed critical and constructive evaluation of our manuscript and are grateful for the acknowledgment of our analytical work. The reviewer's two major points of disagreement are **i) our argument that $\delta^{13}\text{C}$ of crenarchaeol can consistently reflect $\delta^{13}\text{C}$ of seawater DIC via a sufficiently constrained ϵ offset, and ii) our argument that crenarchaeol $\delta^{13}\text{C}$ at our study sites reflects the global $\delta^{13}\text{C}$ record of the marine DIC pool.** We now support our claims about these two points with additional external data, and we have expanded the discussion in the manuscript to address the concerns of Reviewer #1.

We summarize our response to points **i)** and **ii)** here, in addition to the detailed answers below. For point **i)**, we expanded the discussion on carbon isotope fractionation in Thaumarchaeota based on a thorough re-evaluation of literature data and on newly developed models of thaumarchaeal carbon isotope systematics. This discussion includes information from three recent papers from our group, one published (Polik et al., 2018) and two that currently are accepted at *Geochimica et Cosmochimica Acta* (Hurley et al., 2019; Pearson et al., 2019). Collectively, there is no evidence to suggest significant incorporation of exogenous organic carbon (heterotrophy) into thaumarchaeal lipids. Instead, carbon isotope fractionation (ϵ) in Thaumarchaeota is predicted (Pearson et al., 2019) and shown in the modern environment (Hurley et al., 2019) to decrease asymptotically at high $p\text{CO}_2$ concentrations; i.e., become more constant in high- $p\text{CO}_2$ worlds. We present a new figure (Supplementary Fig. 4) calculated for hypothetical PETM conditions that shows how the effect on ϵ is small but significant. Using existing reconstructions and model estimates of Paleocene-Eocene boundary conditions (temperature, salinity, [DIC], $p\text{CO}_2$), we now correct the magnitude of the CIE reconstructed from $\delta^{13}\text{C}_{\text{Cren}}$ for this effect on ϵ , and provide an estimate of the uncertainty (CIE = $-4.0 \pm 0.4\text{‰}$). Further we detail how our previous work indicates that even strong ecosystem changes (stratification, deoxygenation, eutrophication) have negligible effects on $\delta^{13}\text{C}$ -Cren (Polik et al., 2018).

Regarding point **ii)**, we find no quantitative basis for the claim that only the deep ocean could register a global CIE. It is clear from model simulations (e.g., Kirtland Turner and Ridgwell, 2016) that the CIE would equilibrate between the exogenic sub-reservoirs on millennial timescales. Although the equilibration time would depend on the rates, mass and carbon isotopic composition of carbon emissions, this equilibration time would at most be ~ 4 ka, which

is the approximate temporal resolution of our records. Similarly, we argue strongly that our finding of identical magnitudes for the CIE in three different oceanographic regimes would be unlikely to arise by coincidental agreement of various complex local effects; the constant signal is direct evidence that it is global.

27. here 'early paleogene' seems a stretch as it regards the interval between 66 and 34 million years ago. The hyperthermals are within a much more narrow age band.

We understand the concerns raised by Reviewer #1 on our usage of the term "Early Paleogene", as there is no formal definition of this specific term. However, we view the term 'early Paleogene' as appropriate in the context used in our manuscript, as the known hyperthermals span the range between ~65 Ma (Dan-C2 Event) and ~42 Ma (Chron C19r event). This terminology is common when referring to the geological record of hyperthermals of the early Cenozoic (e.g., Tripathi et al., 2003; Sluijs and Dickens, 2012; Westerhold et al., 2012; Hollis et al., 2012; Kirtland Turner et al., 2017; Kozdon et al., 2018a).

30. consider if methane in clathrates can properly be referred to as fossil organic carbon

Production of methane in marine sub-surface sediments and incorporation into clathrates is temporally removed from its potential release during the PETM. As the build-up of methane clathrates in marine sediments must occur on geological timescales to provide a carbon reservoir large enough to contribute to PETM-scale carbon emissions (e.g., Dickens, 2003), we consider the term "fossil" to reasonably reflect the organic carbon in marine methane clathrates.

33/34. Here a unidirectional causal relation is inferred: emissions cause warming. But reference 5 suggested this was the other way around.

Agreed. We removed the directionality, as this is indeed a complicated aspect of climate variability (i.e., lead-lag relationship, cause versus effects, causation and correlation etc.). The statement now reads: "*The most intense hyperthermal, the Paleocene-Eocene Thermal Maximum (PETM), was a ~200 ka global climate perturbation (Westerhold et al., 2018) associated with increased atmospheric CO₂ levels (Schubert and Jahren, 2013; Cui and Schubert, 2017), ocean warming (Dunkley Jones et al., 2013), and ocean acidification (Zachos et al., 2005; Penman et al., 2014).*" (line 28-31)

44-61. This problem was systematically approached by Sluijs and Dickens (your ref 43). From their abstract: 'Individual carbon isotope records differ in shape and magnitude from variations in the global exogenic carbon cycle through changes in (1) the relative abundance of mixed components with different d13C within a measured substrate, (2) isotope fractionation through physiological change, and (3) the isotope composition of the carbon source. All three factors likely influence many early Paleogene d13C records, especially across the PETM and other hyperthermal events.' Particularly because this paper claims to have found the true value of the CIE, these three problems need to be excluded.

We appreciate the recommendation of the reviewer and agree that all potential biases originating from physiological and environmental change need to be addressed to assess the

fidelity of our CIE proxy. As discussed in detail in response to several comments below (e.g., 70-71, 108-109, 165-167) we have systematically assessed the impact of biases from all three categories in our revised manuscript (e.g., category 1: changes in sedimentation and source mixing; category 2: isotope fractionation in Thaumarchaeota; category 3: propagation of the CIE through the exogenic sub-reservoirs).

69. The cited paper (ref 32) reports mean fractionation values of about 19.5 to 19.8. This is an important number but also, The 19.5 – 19.8 values are mean values without uncertainty so here you want to include an uncertainty analysis based on the available data.

We agree that this represents 0.3‰ of uncertainty that should be propagated with the uncertainty of the CIE magnitude from the site-to-site data. More generally, we have addressed this need for better discussion of uncertainty by expanding our discussion of the sensitivity of ϵ to changing $p\text{CO}_2$. However, there is a second important point about ref 32: in that work, the ϵ values reflect the fractionation between DIC and the biphytane sidechains of GDGTs, whereas here we are measuring ϵ relative to GDGTs directly. Pearson et al. (2016) estimated that the difference between GDGTs and biphytanes is 1‰. This means the $\epsilon \approx 19.5$ to 19.8 values are equivalent to $\epsilon \approx 18.5$ to 18.8 in our work. To further constrain ϵ , we have now applied the flux balance model of Pearson et al. (2019) to pre-PETM and peak-PETM boundary conditions (new Supplementary Fig. 4). We find that average ϵ values were 0.6-0.9 permil smaller during the peak PETM compared to the pre-PETM. We have used these values to calculate corrected Cren-based CIE estimates and have propagated the respective uncertainties. Our approach is described in more detail in the revised Methods section.

70-71. Thaumarchaeota predominantly reside at the base of the photic and upper sub-photic zones (ref 37,38). Much evidence, from both modeling and field data suggest that water column structure and productivity changed significantly during the hyperthermals (I presume Carmichael et al., 2017 summarize this evidence but I have not checked). This could affect the result in two ways. 1) it could have affected the DIC-d13C gradient from surface to deep (Sluijs and Dickens issue 3), and 2) it might have affected the archaea thriving in the subsurface; they may have migrated to greater or lesser depth (Sluijs and Dickens issue 3), there could have been a change in communities (Sluijs and Dickens issue 1) or they change their fractionation (Sluijs and Dickens issue 2). If you can address all of these potential errors, you arrive at the local expression of the CIE of DIC; then the step to the global exogenic one needs to be made.

We agree that ecosystem changes could potentially affect thaumarchaeal communities, but based on our recent work on Plio-Pleistocene Mediterranean sapropel events (Polik et al., 2018), we argue that this does not significantly impact values of $\delta^{13}\text{C}_{\text{Cren}}$. In our revised manuscript, we have expanded the discussion of these factors, specifically noting that Thaumarchaeal ecology indicates that their depth habitat should migrate with the depth of maximum remineralization rates at the base of the photic zone and therefore maintain a constant position relative to the $\delta^{13}\text{C}_{\text{DIC}}$ gradient. This assertion is directly supported by our previously published sapropel data (Polik et al., 2018). We want to emphasize that location-specific factors were discussed in the

original manuscript to the extent to which they are known (and are discussed in detail for the Arctic Ocean).

The revised text reads: *“Due to their ecological niche as chemolithoautotrophic ammonia oxidizers, Thaumarchaeota are most abundant in a narrow interval at the base of the photic zone (Church et al., 2010; Santoro et al., 2010). Consequently...the signal should principally reflect basal photic zone $\delta^{13}C_{DIC}$. A point-source origin of $\delta^{13}C_{Cren}$ from the basal photic zone is further supported by $\delta^{13}C_{Cren}$ records of Pleistocene Mediterranean sapropel events S3 and S5 (ref. (Polik et al., 2018)), where significant changes in water column hydrography and changing contributions from deep-water Thaumarchaeota did not lead to significant changes in sediment $\delta^{13}C_{Cren}$. The consistency of the $\delta^{13}C_{Cren}$ data through the sapropel events (including pre-event comparison values) imply that strong changes in ecosystem states, such as eutrophication, habitat depth changes, and anoxic/euxinic conditions do not significantly impact $\delta^{13}C_{Cren}$ values (Polik et al., 2018). This is explained by recognizing that the depth habitat of Thaumarchaeota is linked to oceanographic parameters...thus, the thaumarchaeal population should maintain a similar relative position along the $\delta^{13}C$ gradient...” (Line 196-211)*

81-83. So if this is all true the uncertainty is now between 2,200 and 11,000 Pg. Is that a big improvement?

Our new constraints on the CIE from $\delta^{13}C_{Cren}$ define a tuning target for future modeling as well as a tighter constraint on the upper and lower bounds of carbon emissions. The most important part of our paper is not to define the emission sources, but rather to greatly increase confidence in the value of (and decrease the uncertainty in) the magnitude of the CIE itself. Previous emission models based on lower CIEs of 2-3‰ may have significantly underestimated both mass and carbon isotopic composition of the carbon source. In the revision, we now emphasize that we are agnostic about the specific source and focus instead on why a refined CIE is valuable: *“...Our CIE estimate of $-4.0 \pm 0.4\text{‰}$ is significantly larger than predicted from most marine records and therefore also larger than has been used in most mass balance calculations and carbon cycle models (Dickens et al., 1995; Zeebe et al., 2009; McInerney and Wing, 2011a; Gutjahr et al., 2017). ... The range of $\delta^{13}C_{Cren}$ -based emissions estimates is not only narrower than previous mass balance estimates but also more precise, with an uncertainty of ~10% compared to the much larger uncertainties resulting from the wide scatter of other substrate-specific CIEs (McInerney and Wing, 2011a) (Fig. 3A)...” (line 334-344)*

93-95. Awkward phrasing; suggests that the Lomonosov Ridge was a restricted basin.

Agreed. We rephrased the sentence: *“The Arctic Ocean site was located on the submerged northern flank of the Lomonosov Ridge⁴².” (line 106-107).*

96-98. This is different than most previous approaches, where the difference between average so-called ‘body’ CIE values and pre-CIE values is taken. Why choose the minimum value here rather than a mean of multiple analyses with a proper statistical measure of uncertainty? It is indicated that the reported uncertainty includes analytical uncertainty but not how this stratigraphic issue is managed.

We agree with Reviewer #1 that mean pre-PETM and mean peak PETM $\delta^{13}\text{C}$ values provide the most conservative CIE estimate. However, there are no accepted guidelines and the methodology to calculate CIEs is commonly not reported. We revised our manuscript to now use a reproducible and statistically objective approach to constraining the mean value of the maximum CIE amplitude. We employ the “*rampfit*” method developed in Mudelsee (2000) to objectively calculate the onset and end of the PETM transition (based on least-square regression; Mudelsee, 2000). This allows us to obtain robust estimates of mean peak (“body”) $\delta^{13}\text{C}$ - and pre-PETM $\delta^{13}\text{C}$ values that provide the basis for improved CIE calculations in our revised manuscript. We note, however, that the Arctic data are not continuous and cannot be estimated using *rampfit*; therefore, we had to retain the minimum value approach for this record.

The results are as follows “*The intervals of the pre- and peak-PETM were determined through ramp function fitting (Mudelsee, 2000) (rampfit; Supplementary Fig. 1). For the Arctic Ocean record, owing to the coring gaps, we report the CIE as the difference between the pre-PETM $\delta^{13}\text{C}_{\text{cren}}$ mean and the lowest $\delta^{13}\text{C}_{\text{cren}}$ value during the PETM. We find a CIE_{cren} of $-3.12 \pm 0.20\text{‰}$ for the Ancora site and $-3.28 \pm 0.21\text{‰}$ for the Tasman Sea (Fig. 1). The CIE_{cren} estimate for the Arctic Ocean (minimum-value approach) is $-3.31 \pm 0.23\text{‰}$.” (line 121-127).*

Supplementary Fig. 1 shows the fitted ramp functions (*rampfit*) not only for our $\delta^{13}\text{C}_{\text{cren}}$ dataset but also for existing proxy data records (foraminifera, TOC, other biomarkers) for each of our three study sites. It hence allows a robust comparison between our data and previously reported CIE magnitudes for the three study sites (Cramer and Kent, 2005a; Sluijs et al., 2006; Pagani et al., 2006; John et al., 2008; Sluijs et al., 2011; Sluijs et al., 2018).

101-102. Is this just the average of the three numbers with the propagated uncertainties?

Yes, this is the average CIE with propagated uncertainties. We have described the uncertainty propagation in the revised methods section and rephrased this sentence: “These estimates are all consistent within 1σ uncertainties, with an average CIE_{cren} of $-3.24 \pm 0.37\text{‰}$.” (line 127-128)

106-108. It is relevant here to compare the Arctic result to previous work, notably that of the short chain alkanes of Pagani et al., (2006, Nature; currently not cited) and that of the synthetic marine organic carbon record from ref 43?

We agree with the reviewer and have added comparisons with both substrates to the revised manuscript, as outlined below in the response to the reviewer’s comment on Figure 1.

108-109. At the New Jersey shelf, the only site studied here with sufficiently preserved carbonates, benthic foraminifera indicate a CIE of $-3.4 \pm 0.14\text{‰}$ (ref. 44). It is generally accepted that seawater carbonate chemistry affects carbon isotope ratios in foraminifera (e.g., Spero et al., 1997 and follow-up work). This has been explored for the PETM by ref 57, at least for planktic foraminifera, and found to dampen the CIE. If so, why is the local CIE in archaeal lipids not larger?

We agree that several factors may have dampened the CIE in foraminifera, as shown by the dampened global median CIE in foraminifera (-2.6‰ , Fig. 3; (McInerney and Wing, 2011b))

relative to crenarchaeol (~-4‰). However, the expression of dampening effects such as post-depositional carbonate dissolution is location dependent, and a variety of other processes can cause disequilibrium effects between foraminiferal $\delta^{13}\text{C}$ and $\delta^{13}\text{C}$ -DIC during calcification (Ravelo and Fairbanks, 1995; Spero and Lea, 1996; Spero et al., 1997; Kohfeld et al., 2000). In fact, dampening effects through dissolution and CO_2 - changes during calcification (biasing the CIE towards smaller values) may be cancelled out or counteracted by changes in metabolic rates and carbon uptake due to seawater temperature changes, and the $\delta^{13}\text{C}$ of the food source (i.e., both biasing the CIE towards larger values). We have modified the manuscript as follows:

“At the New Jersey shelf, the only site studied here with well-preserved (“glassy”) foraminifera, benthic foraminifera indicate a CIE of $-3.5 \pm 0.3\text{‰}$ (Ancora) (Cramer and Kent, 2005b) and $-3.8 \pm 0.4\text{‰}$ (Bass River (John et al., 2008); Supplementary Fig. 1), while the CIE derived from planktic foraminifera (Bass River) is somewhat more variable (3.6-4.3‰) (John et al., 2008). Along with other CIE estimates from well-preserved foraminifera from the South Atlantic ($-3.4 \pm 0.2\text{‰}$, paleo-water depth of ~1500 m; Supplementary Fig. 1) (McCarren et al., 2008) and the North Atlantic ($-3.3 \pm 0.2\text{‰}$; Supplementary Fig. 1) (Cramer and Kent, 2005b), these estimates are consistent with our $\delta^{13}\text{C}_{\text{Cren}}$ -based CIE_{DIC} estimate, emphasizing that the global median CIE of planktic foraminiferal records (McInerney and Wing, 2011b) of -2.6‰ is biased by generally poor carbonate preservation. ...” (Line 304-313)

Figure 1. - here the $\delta^{13}\text{C}$ -cren results are compared to $\delta^{13}\text{C}$ -TOC. Why? For both the Arctic Ocean and New Jersey sites, $\delta^{13}\text{C}$ records of more specific marine substrates have been generated that allow for a better comparison.

We agree with the reviewer that comparisons with more specific substrates are valuable and have included these in the revised manuscript. Specifically, we included a comparison with data from short-chain *n*-alkanes (Pagani et al., 2006), as well as dinoflagellates and foraminifera from New Jersey (John et al., 2008; Sluijs et al., 2018). In addition, we expanded the original discussion on the marine organic carbon record from the Arctic Ocean. We also retain the comparison with $\delta^{13}\text{C}$ -TOC, as it is the only substrate for which data are available at all three sites.

- I assume that the $\delta^{13}\text{C}$ -cren data were generated on the same extracts used for TEX₈₆ analyses by the original papers but this is mentioned nowhere.

TEX₈₆ and $\delta^{13}\text{C}_{\text{cren}}$ data were generated from the same extracts for the New Jersey site. For the Tasman Sea and Arctic Ocean sites, we used adjacent sediment samples from the same cores. This is now clarified in the methods section: *“The TLEs were split into two aliquots for i) for purification and carbon stable isotopic composition of crenarchaeol, and ii) analysis of lipid abundances for determination of TEX₈₆ and BIT.”* And: *“Relative abundances of GDGTs (Supplementary Data Table S3) from site ODP 174AX were determined from aliquots of the same TLE used for crenarchaeol isotopic analyses [...] Published TEX₈₆ and BIT data from adjacent samples were used for the Arctic Ocean (Sluijs et al., 2006) and Tasman Sea records (Sluijs et al., 2011)”* (Line 410-412, 431-436).

128. BIT is considered a tracer of soil organic matter as far as I know.

128-130. Strange sentence

This sentence was missing a word and was revised to specify soil organic matter. Revised: “*To assess the influence of terrigenous input of crenarchaeol on $\delta^{13}C_{Cren}$ records of BIT (branched-over-isoprenoid tetraether index), a semi-quantitative tracer for input of soil organic matter (Hopmans et al., 2004), [...]*” (Line 110-112)

128-144. This was all extensively discussed in previous work; why repeat here (in lines 128-139 even without referring to that work)? It is in the results section, which makes it extra weird because it regards data generated in previous work (I think except for this New Jersey record).

We agree with the reviewer and have revised the text by i) clarifying the origin of data in the results and methods sections, ii) deemphasizing the data description that relates to literature data. However, it is important to contextualize the new $\delta^{13}C_{Cren}$ and TEX86/BIT data with existing data from the same cores. Therefore, a brief description of the literature is necessary.

157. If it is 100% certain that Thaumarchaeota fix ONLY bicarbonate, that should be specified here. If it is uncertain if also other carbon sources may be incorporated (CO₂ or some degree of heterotrophy), that should be mentioned here too.

Yes, the pathway employed by Thaumarchaeota is specific for bicarbonate. We have added a reference to Berg et al. (2007) to support this statement. We also have further refined the discussion on potential heterotrophy as indicated below.

161. Carbon isotopic fractionation in thaumarchaeota is understudied and the constraints may be insufficient to make the claims that are presented in this paper. Line 159-160, for example, indicates that the authors don't expect differences between clades but with the present data, there is no way to evaluate that claim. In addition, I am missing a critical evaluation of potential other factors influencing $\delta^{13}C$ fractionation. It is important to note that ref 32 studied if the biphytanes are a decent proxy for bulk biomass in the species *Nitrosopumilus maritimus*, not if the fractionation between DIC and biomass is constant. Ref 32 explicitly expects changes in fractionation as a function of temperature and pH. These factors are currently ignored while both factors vary between study sites and change dramatically during the PETM.

We agree that the carbon isotopic fractionation in Thaumarchaeota needed further discussion in our manuscript, but there is considerably more information available than the reviewer suggests. Published isotope data on two organisms using the 3-hydroxypropionate-4-hydroxybutyrate cycle (*Nitrosopumilus maritimus* and *Metallosphaera sedula*), as well as new environmental data (Hurley et al., 2019) and flux-balance models (Pearson et al., 2019) from our group allow a mechanistic investigation of carbon fixation in Archaea using this cycle. Specifically, we can show that the carbon isotope effect in Thaumarchaeota originates from diffusive uptake of CO₂ and is therefore sensitive to pCO₂. We now quantify the predicted pCO₂ sensitivity for pre and peak PETM boundary conditions (new Supplementary Fig. 4) and have corrected our $\delta^{13}C_{Cren}$ data for this effect when calculating the CIE. (line 186-195)

164. Isn't there earlier work that already showed this than ref 47?

Ref 47 is the most comprehensive study on this topic. There is earlier, less conclusive work on the topic which was not cited due to the reference limit. We have asked the editor to expand the limit and included a reference to the earlier work by Wuchter et al. (2005).

165-167. I disagree with this statement as it assumes that things don't change at the PETM. They do in a relevant way. For both New Jersey and the Arctic, big changes in the hydrological regime have been recorded and that should affect water column structure and productivity and that has been described in several papers for both regions. It is therefore unlikely that the $\delta^{13}\text{C}$ gradient of DIC and the depth of origin of the isoprenoid GDGTs remained the same. This is important because it directly affects the main conclusion.

We agree that this argument needs to be laid out in more detail and have expanded this section. Importantly, we discuss our recent work from the Mediterranean Sea, where sapropels record ecosystem changes similar to those in the Arctic Ocean during the PETM but show no evidence for changes in $\delta^{13}\text{C}$ -Cren due to water column stratification, deoxygenation, or eutrophication (Polik et al., 2018). For the other two sites, we focus on commonalities in ecological change and their impact on thaumarchaeal ecology and water column structure.

168-170. This is a major issue and I disagree with the way this is disregarded as a potential problem. All micropaleontological studies indicate a massive change in ecosystems during the PETM. Here it is stated that a significant portion of the GDGT carbon atoms may derive from some food source. However, during the PETM the $\delta^{13}\text{C}$ of that food source will not change with the same magnitude as the $\delta^{13}\text{C}$ of DIC (that is the implicit assumption, not the $\delta^{13}\text{C}$ values that are given here). This fact is actually the motivation of this study. Even if the uncertainty is 1‰ or less, this probably implies that this uncertainty must be propagated in the magnitude of the CIE that is reconstructed. Why is it not?

170-173. Indeed, so if this is unknown, how can we really tell that you're correct? Seems very opportunistic.

The premise of the reviewer's statements is that the archaea are affected by heterotrophic consumption of "food". However, as we now argue more clearly in the paper, previous estimates of thaumarchaeal heterotrophy likely resulted from a variety of erroneous interpretations and assumptions. Presently there is no definitive evidence for incorporation of organic carbon into thaumarchaeal lipids. Please see extended discussion in Lines 218-230.

174. Strange sentence. What does 'its' refer to?

Rephrased to: "Systematic offsets between $\delta^{13}\text{C}_{\text{Cren}}$ values at our three study sites additionally corroborate an autotrophic origin of crenarchaeol." (Line 231-232)

183. Ref 35 reports higher values; important here to include the full uncertainty as mentioned above.

We have revised this section by including propagated uncertainties in our DIC estimates.

186. What is a 'basinal environment'?

Revised: "...accumulation of ^{13}C -depleted DIC in the chemocline and anoxic zone of the Arctic..."

187. Are there indications that euxinic conditions prevailed in the Arctic prior to the PETM?

In the available literature, there is currently no evidence for euxinia in the Arctic Ocean immediately preceding the PETM. However, anoxic or euxinic conditions cannot be excluded as the pre-PETM hydrography is unconstrained due to a lack of deep marine records.

188-189. Why would this lead to an overestimate of the CIE? As BIT drops during the PETM shouldn't it lead to an underestimate?

The reviewer is correct, it may lead to a dampening of the CIE. However, crenarchaeol is not abundant in soils and thus any influence may be minimal. We have revised this paragraph for clarity: *"The BIT record at our deep Arctic Ocean site indicates significant soil input only during the pre-PETM interval (Supplementary Fig. 2). However, it seems unlikely that terrigenous input could significantly impact marine $\delta^{13}C_{Cren}$ values as crenarchaeol is a minor or trace component in soils compared to other GDGTs (Weijers et al., 2010). This view is supported by the consistency of the CIE_{Cren} between sites throughout the analyzed time interval regardless of the magnitude of the local BIT value (Supplementary Fig. 2)."* Lines 249-255.

191-200. Consistency between the sites cannot be used as an argument. All potential confounding factors should be dealt with at the individual sites. Then it should be evaluated if consistency between 3 sites gives you the global exogenic CIE or not.

We agree that this argument needs to be laid out in more detail and have re-framed the discussion on BIT values. We strongly disagree however, that consistency cannot be used as an argument. Consistency between sites underlines our argument for low biases based on locality. Additionally, the potential site-dependent biases are discussed already in the manuscript. The Ancora and Tasman Sea sites have a similar setting and thus for some aspects we focus on an in-depth discussion of the Arctic Ocean record. However, even given the strong oceanographic differences, there is no evidence for anomalous $\delta^{13}C_{Cren}$ values in the Arctic record.

204-206. If so, than this affects the potential heterotrophic contribution to the GDGTs.

As detailed above, we argue there is no heterotrophy.

209-212. I don't understand this; both factors have been shown to lead to larger CIEs in organic matter (see, e.g., plant community shift study of Schouten et al., 2007 EPSL)

We have removed this statement as it is not central to the manuscript and because biases were already discussed earlier in the text.

Figure 3. - the bulk marine organic matter data; isn't that rather TOC from marine sediments (so really a mixture of marine and terrestrial organic carbon)?

Label changed to "Marine sediment TOC".

237-243. Not if a pH effect is added to the foram records

We agree that changes in pH could impact foraminiferal $\delta^{13}\text{C}$ records. However, there are multiple additional effects that may positively or negatively bias foraminiferal $\delta^{13}\text{C}$ values, as outlined above in our response to the comment on 108-109.

241. The benthics from ref 26 are heavily dissolved and recrystallized. Far from 'glassy'.

We agree and have modified the statement accordingly: *"At the New Jersey shelf, the only site studied here with well-preserved ("glassy") foraminifera [...]"* (Line 304-305)

246-247. Why is this a strong point?

Comprehensive CIE reconstructions rely on global datasets, which in large parts of the ocean is hampered by carbonate dissolution. $\delta^{13}\text{C}_{\text{Cren}}$ data may come from areas that may not have well-preserved carbonate. In order to streamline the discussion, we have removed this statement. However, the essence of this statement is preserved in our final statement in the discussion: *"The diagenetic stability and potential for high-resolution records of archaeal lipids suggest that $\delta^{13}\text{C}_{\text{Cren}}$ is a promising proxy, particularly for reconstructing transient CIEs that exhibit poor carbonate preservation, such as during Mesozoic Oceanic Anoxic Events and early Cenozoic hyperthermals."* Line 396-399.

248-249. Why is the CIE not compared with planktic or thermocline-dwelling foraminifera at New Jersey? They may be closer to the position of the archaea than the benthics. Then you don't have to assume that the $\delta^{13}\text{C}$ depth gradient remained identical during the PETM (which it probably didn't).

We have revised the discussion to include comparison with planktic foraminifera from New Jersey (John et al., 2008), but this comparison did not affect our findings.

260 Most of the recent modeling regarding the mass of injected carbon during the PETM has focused on the total mass of carbonate dissolution (Panchuk, Zeebe, Ridgwell, Cui etc). This has constrained the mass a little better than the wide range suggested here.

We agree that these models have yielded further constraints on carbon emissions. However, it is important to note that these models are at odds, with Zeebe et al. proposing an upper bound of 3,000 Pg C while Panchuk et al. propose an upper limit of 7000 Pg C and Gutjahr et al. suggest over 10,000 Pg C. In order to discuss these results in more detail, we have added the following statement: *"Carbon cycle modeling based on observed changes in ocean carbonate saturation and ocean pH yield CIE-independent, but diverging estimates of ~3000-7000 Pg C (refs. (Panchuk et al., 2008; Zeebe et al., 2009)) and >10,000 Pg C (ref. (Gutjahr et al., 2017)), respectively. However, reconstructions of the isotopic composition of the carbon source from these models still depend on assumptions about the CIE magnitude (Kirtland Turner and Ridgwell, 2016, p.201)."* (Line 55-59)

261. Is the number 34,000 correct? Seems small (smaller than at present)

This was corrected to 40,000 Pg in response to a comment by Reviewer #2.

266. It depends on a little more than that if methane hydrates were involved (Dickens, 2011 CP) given the time scale of steady state disruption.

We agree that multiple uncertainties limit PETM emission estimates. However, we have condensed this part of the discussion and this statement was removed.

271 and further. This exercise is identical to that presented in multiple papers.

We have streamlined this section to avoid reiteration of previous studies.

277. Please do read the Dickens papers carefully (1995 Paleoceanography and 2011 CP) for a more nuanced statement on this. The methane hydrate hypothesis needs warming to release the methane as a positive feedback. It was never an explanation for the warming until Pagani et al (2006) thought it was and subsequently excluded it.

We agree with the reviewer. Following the restructuring of this section based on other reviewer comments this statement is not retained in the revised manuscript.

302-304. I don't think any of the papers that have suggested pre-CIE warming have suggested this for your study sites. Your Arctic record has no recovery for the onset of the CIE; your southern ocean record has very low time resolution; your new jersey record may well be incomplete at the onset. So how can you really study this question?

We agree that these sites are not ideal to study lead-lag relationships of temperature and CIE. We have rephrased this section.

305. High sampling resolution but the thickness of the CIE there, following your figure 1, is less than a meter and the onset perhaps condensed due to sea level rise (ref. 40). So that is why a large community has focused on several New Jersey shelf sites, where sedimentation rates are more than 10 times higher.

We agree and have rephrased this section to rather highlight the utility of the approach.

308-310 I fundamentally agree with this point. To have full added value, though, sites where lead/lag relations have been identified should be tested.

We agree that lead/lag relationships should be tested using multiple proxy couples. In order to add to this statement, we have added a reference to the recently published work by Frieling et al. (2019) on pre-CIE warming.

313-314. The Arctic Ocean site has no recovery across the onset; the top of Core 32X represents caved sediments (Pagani et al., 2006). Events similar to Bowen et al's POEs actually have been found in other New Jersey cores.

We agree and have revised this section to highlight the general application in locations other than the Arctic. The revised text reads: *"The synchronous change in $\delta^{13}C_{Cren}$ and TEX_{86} at our study sites (Fig. 2) suggests that any warming preceding the CIE must have occurred no more than 3-4 ka prior, consistent with previous estimates from the New Jersey shelf (Sluijs et al., 2007; Frieling et al., 2019), the equatorial Atlantic (Frieling et al., 2019), and the Southern Ocean*

(Frieling et al., 2019). *The coupled $\delta^{13}C_{Cren}$ -TEX₈₆ approach described here could be instrumental for determining high-resolution lead-lag relationships of temperature and CIE during the PETM and other climate events.* (Line 374-379). We could not find any published reports on POEs specifically from the New Jersey shelf.

329. but the present CIE has not been mixed with the entire global exogenic carbon pool, notably the deep ocean! And based on what is the termination of the present perturbation more similar to POE's than to the PETM? If the POEs are truly global carbon cycle perturbation, we really do not understand how they ended so quickly. See papers by Archer on the long-term legacy of anthropogenic carbon.

We agree. We have removed this statement and have significantly condensed the discussion on the POE to retain the focus of the manuscript on the PETM.

336-337. I disagree. The present records suggest instantaneous recovery from POE's while, e.g., ETM2 recovery was in the order of 40 kyr (e.g., Stap et al., 2009 Paleoclimatology).

We agree and have removed the reference to other Paleogene hyperthermals.

339-340. It is probably important to read the literature on PETM deposition along the New Jersey shelf in a bit more detail. It is well possible that the studied core is incomplete at the onset. Other cores show pre-CIE warming.

We agree that other cores show pre-PETM warming. The pre-PETM cooling that we were referring to here was the cooling after the POE and before pre-PETM warming. We have deleted the statement since it may be perceived as misleading.

399-400. Do TEX₈₆ values correspond well to the previously studied PETM sections from this region?

TEX₈₆ values for the Ancora record correspond closely to previous data from the same region at Bass River, showing values of ~0.75 in the Paleogene and a rise to ~0.9-0.95 during the peak CIE. We have added this information: *"Concomitant to the CIE, TEX₈₆ values increase by ~0.2 in the Ancora record, indicating ocean warming, and return to pre-CIE levels after the PETM. A similar pattern was observed in the Bass River (Sluijs et al., 2007) and Tasman Sea records (Sluijs et al., 2011) (Fig. 2)."* (Line 157-159)

Reviewer #2

I have read through the manuscript (MS) by Elling and colleagues several times. The MS tangentially addresses several issues in the geosciences, including one of the top current and cool problems facing our community, namely how can massive amounts of carbon enter the ocean and atmosphere really fast in the past, such as across the Paleocene Eocene Thermal Maximum (PETM). This was a geologically brief time of major warming and environmental change, and serves as perhaps our best past analog for anthropogenic changes.

The authors measure the $\delta^{13}C$ of crenarchaeol at three sites across the PETM. This is novel and sort of cool. They show that the CIE excursion in these records is about -3.5 per mil.

Per background, and as noted at the start of their MS, a prominent negative CIE manifests in all carbon-bearing phases across the PETM. For 23 years, this has suggested as representing a massive input of ^{13}C -depleted carbon into the exogenic carbon cycle (ocean, atmosphere and biomass) (Dickens et al., 1995). The source of this carbon has remained somewhat controversial, in part because the magnitude of the CIE varies between locations and substrates analyzed.

Several papers have argued that the CIE expressed in certain records represent global changes in the carbon isotopic composition of the entire exogenic carbon cycle. This has never made sense to me, but remains circa 2018 a “debatable” view. We thus have papers invoking mechanisms to explain a <3.0 per mil $\delta^{13}\text{C}$ excursion to >5.0 per mil $\delta^{13}\text{C}$ excursion

I have mixed views of the MS. On the one hand, the data is novel and interesting; on the other hand I personally learned nothing new in the big picture. But this is perhaps because I have worked on the problem for over two decades and it always has been clear to me that the CIE must be around 3.0 per mil, and deviations from this value in various substrates represent one of several processes (Sluijs and Dickens, 2012). There are also some awkward issues and numerous minor items. I note these on an annotated document and below.

I think the MS needs several significant modifications before publication, but with amendments could make a nice contribution.

Sincerely,
Gerald (Jerry) Dickens

Issues elaborated on from annotated manuscript

(A) The manuscript does not fully capture and explain key issues. The biases and discrepancies lie in individual CIE measurements and how they relate to changes in the exogenic carbon cycle. This issue arises in multiple places and has to be amended.

We have rephrased several statements in the introduction and discussion sections to better distinguish between individual (location-specific) CIE estimates vs. the CIE expressed in the total exogenic carbon cycle. We have also clarified how the CIE in crenarchaeol relates to and should reflect the total exogenic CIE. Following model estimates of Kirtland Turner and Ridgwell (2016), the CIE can be expected to be equilibrated among the different exogenic sub-reservoirs within the temporal resolution (3-4 kyr) of our $\delta^{13}\text{C}$ -Cren records.

(B) There is a nominally 19 per mil fractionation between DIC and crenarchaeol (Figure 1 and Line 69). However, it's not crystal clear in the text (Lines 69+, and 155+) how and why this fractionation occurs. Note here that DIC includes both HCO_3^- and CO_3^{2-} and the pH across the PETM changed (as well as likely the relative proportion of HCO_3^- and CO_3^{2-}). So, why would the fractionation remain constant? This is confounded by the issue noted on Line 169 that there be mixing. Albeit $<$ than 1 per mil, this is huge in the context of the MS goals. Even assuming that the fractionation remained constant, why would the $\delta^{13}\text{C}$ of the lower photic maintain a constant offset to the exogenic carbon cycle?

This is sort of addressed from an observational point later (Line 194+).

We have expanded the discussion on fractionation and potential biases of $\delta^{13}\text{C}$ -Cren in response to similar questions raised by Reviewer #1. Specifically, we have included new discussion and additional citations about carbon isotope fractionation in Thaumarchaeota. In the revised discussion we also specify that heterotrophy and/or ecosystem changes are not confounding factors and that the only major variable affecting fractionation would be changing CO_2 levels. The changes in CO_2 levels under Paleocene-Eocene conditions have been used to apply a correction factor to the fractionation factor and thus also to the CIE estimate, including associated propagation of uncertainty.

(C) I do not understand Figure 2 in the details. The relationship between theoretical carbon inputs and $\delta^{13}\text{C}$ has been discussed many times since 1995, although I think Pagani et al. (2006) were the first to put into such a conceptual diagram. Obviously, the relationship depends on the mass and $\delta^{13}\text{C}$ of exogenic carbon cycle. The $p\text{CO}_2$ relationship (right axis) is not so simple and depends on multiple factors. More crucially, I do not understand the points, as many of these references discuss multiple possibilities.

Figure 3 [not Fig. 2] is a compilation of studies that have reported both a CIE value and an emission estimate. This figure visualizes the parameter space of emissions that are compatible with our $\delta^{13}\text{C}$ -Cren CIE estimate, as well as previous estimates. Each number/symbol represents a distinct emission scenario and the substrate type on which the CIE was measured. For studies that have reported multiple scenarios, these scenarios are distinguished by individual numbers (this point is clarified in the revised figure caption). The full list of scenarios and parameters can be found in Supplementary Data Table S1. We agree that the $p\text{CO}_2$ relationship is complex and have therefore removed the right axis.

(D) It seems that the authors also measured $\delta^{13}\text{C}$ of TOC, but this is not really emphasized in the Introduction and results. Indeed, until I read the methods and re-read the caption to Figure, I assumed that these values came from other works. Here it important to note that $\delta^{13}\text{C}$ of TOC and other phases (carbonate) have been measured at all three sites. But a proper comparison is missing.

We have re-organized the results section and the last paragraph of the introduction to more clearly outline which data were generated for this study (and which literature data were used).

(E) There is a lot of “arm-waving” in the last fourth of the paper (Line 271+) on topics that are not solved in the current MS but have been discussed extensively in the literature. I would much prefer if the authors expanded upon the above issues and toned down the speculation and rehashing at the end. I did not learn anything new here.

Following the reviewer’s suggestion, we have toned down speculative and repetitious aspects in the last part of the discussion and have expanded the discussion on the other aspects highlighted by the reviewers (local vs. global CIE, potential biases in $\delta^{13}\text{C}$ -Cren values).

Comments transcribed from the reviewer attachment:

Line 15: corroborate - word choice

Changed to “reflects”: *“A negative carbon isotope excursion (CIE) recorded in terrestrial and marine archives reflects massive carbon emissions ...”* (Line 14-15)

Line 17: Not sure what this means. The issue lies in the interpretation of how the CIE is expressed in various phases. See letter.

We agree that the interpretation of CIE expression among the various phases is a key issue. To more clearly state this, we have rephrased the sentence to *“Yet, discrepancies in current CIE estimates from different sample types lead to substantial uncertainties in the source, scale, and timing of carbon emissions.”* (Line 16-18)

Line 21: This is not necessarily correct, and already mixes measurement and interpretation.

We have revised the abstract to emphasize that the results (-4.0 ± 0.4 ‰ CIE) enable emissions estimates to be refined, but we now choose to remain agnostic about the nature of those emissions.

Line 30: This is problematic as there remains ZERO evidence that the perturbation was triggered by the carbon input as expressed by the CIE. Indeed, multiple explanations for the CIE imply that environmental changes caused, at least in part, the carbon emissions.

We agree and have removed the causality from this paragraph. Rephrased text: *“The most intense hyperthermal, the Paleocene-Eocene Thermal Maximum (PETM), was a ~200 ka global climate perturbation (Westerhold et al., 2018) associated with increased atmospheric CO₂ levels (Schubert and Jahren, 2013; Cui and Schubert, 2017), ocean warming (Dunkley Jones et al., 2013), and ocean acidification (Penman et al., 2014).”* (Line 28-31)

Line 34: As noted above, the cause/effect relationship is by no means obvious.

The text was rephrased accordingly. See response to comment on Line 30 above.

Line 41: Probably should re-read and reference original references on the topic.

We have included additional references (Dickens et al., 1995; Kump and Arthur, 1999).

Line 44: This needs rewriting as it is for mass balance estimates of the CIE.

Rephrased to: *“The magnitude of the CIE can in principle be reconstructed through $\delta^{13}\text{C}$ measurements of carbon-bearing substrates in geologic records, and is thus commonly used for mass balance calculations of PETM emission (Dickens et al., 1995; Higgins and Schrag, 2006; McInerney and Wing, 2011b) (compiled in Supplementary Data Table S1) ...”* (Line 40-42)

Line 45: Should be the exogenic carbon cycle, as it needs to also include the terrestrial biomass

Changed to *“exogenic carbon reservoir”*. (Line 45)

Line 47: But these are not estimates for the magnitude of change in the exogenic carbon cycle; these are measurements of the CIE in various phases.

We agree and have re-phrased this and other introductory paragraphs to distinguish between the CIE measured in different substrates versus the CIE inferred for the exogenic carbon reservoir.

Line 50: Enhanced volcanism makes no sense; the citation refers to thermogenic methane. It probably should be "greatly enhanced seafloor methane emissions from gas hydrate dissociation or heating of organic carbon21 ..."

Rephrased to *"...a wide array of possible carbon sources, including enhanced methane emissions from hydrate dissociation (Dickens et al., 1995), geothermal heating of organic carbon (Svensen et al., 2004), and oxidation of permafrost or other generic organic carbon sources (Higgins and Schrag, 2006; DeConto et al., 2012)."* (Line 53-55)

Line 54: It is really multiple reasons including diagenesis and changing mixtures of components, changing fractionation and changing local conditions. Ref 18 does not really discuss this, but Ref 43 very much highlights this.

We agree and have rephrased the paragraph to highlight the complexities associated with CIE estimates: *"The discrepancy in the magnitude of the CIE between sample types and across sampling locations likely originates from changes in local and biological factors across the PETM that influence $\delta^{13}C$ values of the substrate. These factors include diagenetic overprints, source heterogeneity, and vital effects of the source organisms (Dickens, 2011; McInerney and Wing, 2011b; Sluijs and Dickens, 2012)."* (Line 60-63)

Line 57: What is similarly observed in marine sediments? Not clear.

Rephrased to: *"These factors similarly affect the carbon isotopic composition of marine phytoplankton biomass (Laws et al., 1995; Popp et al., 1998) and thus bias CIE reconstructions from bulk sediment organic matter towards larger CIEs, in addition to offsets resulting from varying admixtures of terrestrial organic matter into marine sediments (Sluijs and Dickens, 2012)."* (Line 66-69)

Line 71: And here lies a potential problem (See text, B)

As described in the response to comment B, we have greatly expanded the discussion to explain why ecosystem changes should have minimal impact on $\delta^{13}C_{Cren}$.

Line 76: There needs to be a comment that the details explained below. When I first read was like what is this?

Rephrased to: *"Here we apply spooling-wire microcombustion–isotope ratio mass spectrometry of $\delta^{13}C_{Cren}$ (described in detail in the Methods section and in ref. (Pearson et al., 2016))..."* (Line 86-87)

Line 77: Not crystal clear what was measured in this study and what was measured before.

We have re-organized the results section and clarified the use of original and literature data in the first part of the results as well as in the methods section.

Line 82: This needs rewording because the huge assumption is that the $\delta^{13}\text{C}_{\text{Cren}}$ represents the change in the exogenic carbon cycle. Probably also should rearrange the numbers as, for example 3,500-11,000 relates to -46 to -16.

We agree and have re-phrased this section to outline how $\delta^{13}\text{C}_{\text{Cren}}$ relates to the CIE in the global exogenic carbon reservoir. Further, we have removed the emission estimates from this section. Revised text: *“We argue that under PETM boundary conditions ($\text{pH} \sim 7.5$, ref. (Penman et al., 2014); $[\text{DIC}] \geq 2 \text{ mM}$) $\delta^{13}\text{C}_{\text{Cren}}$ is offset by a predictable fractionation factor from $\delta^{13}\text{C}_{\text{DIC}}$ (refs. (Hurley et al., 2019; Pearson et al., 2019)) and that the change in $\delta^{13}\text{C}_{\text{Cren}}$ during the PETM is representative of the CIE in the exogenic carbon reservoir.”* (Line 90-93)

Line 83: I am not a big fan of the last sentence here as it is presenting the results and potential significance before the results after already done once before.

Sentence was removed to avoid repetition.

Line 90: What core?

Site Ancora has no core designations other than Hole A/B. This information is now included: *“We sampled the Paleocene-Eocene boundary in three marine sediment cores (Fig. 1) retrieved during expeditions ODP 174AX (Ancora site, Hole A/B, New Jersey shelf), ODP 189 (Hole 1172D, core 15R section 4-5, Tasman Sea), and IODP 302 (Hole M0004A, cores 28X-32X, Arctic Ocean).”* (Line 101-104)

Line 93: Add a bit more on location (e.g., north-west Atlantic; south-west Pacific)

Changed as suggested: *“During the Paleocene and Eocene, the New Jersey shelf and Tasman Sea sites were located on continental margins of the north-west Atlantic and south-west Pacific Oceans, respectively (Sluijs et al., 2011; Stassen et al., 2012).”* (Line 104-106)

Line 95: Well, the ridge did not form the basin.

This sentence was rephrased to: *“The Arctic Ocean site was located on the submerged northern flank of the Lomonosov Ridge⁴²”* (Line 106-107).

Line 101: Change "all" to "three"

Changed as suggested.

Line 113: I think the PETM is defined as the start of the CIE excursion not as the main drop, but maybe things have changed. If right, the PE boundary is shown incorrectly.

We corrected this issue in all figures, with the PE boundary now marked at the start of the CIE.

Line 181: This raises another potential problem, namely that the circulation may have changed during the PETM.

As discussed in the main text, circulation may have reverted during the PETM. However, the overturning rate would have been faster than the temporal resolution of our records (3-4 kyr),

indicating that any changes in $\delta^{13}\text{C}_{\text{DIC}}$ would have been mixed across the ocean and thus rendering $\delta^{13}\text{C}_{\text{Cren}}$ a recorder of longer-term ocean $\delta^{13}\text{C}$ change.

Line 202: No. Bulk marine carbonate at open ocean sites is usually around 3 per mil, so are benthic foraminifera and "thermocline dwelling" foraminifera. It is the planktic foraminifera, which likely hosted photosymbionts, that often give >4 per mil excursions.

In this paragraph, we are referring to "bulk marine organic matter", not "bulk marine carbonate". We acknowledge that photosymbionts and other vital effects influence the CIE in planktic foraminifera. We discuss this in lines 229-236 of the original manuscript.

Line 220: I do not understand this figure (see letter)

As stated in the response to the comment in the letter, we have clarified the figure caption.

Line 224: This is not the correct way to "model" the problem (see Dickens, 2001). This is because the carbon input has to impact also the biomass and atmosphere. So, it should be about 40000 Pg with a $\delta^{13}\text{C}$ of -2.5 permil with the marine components being about 34000 Pg with about 0 per mil.

Following the reviewer's recommendation, we have corrected all calculations by using 40,000 Pg with a $\delta^{13}\text{C}$ of -2.5 permil for the pre-PETM total exogenous carbon reservoir.

Line 238: benthic foraminifera. I am also not sure if monospecific. Clearly monospecific planktic foraminifera generally do not support this (in my opinion because of changes related to photosymbionts).

We generally agree. However, some planktic *foraminiferal records do agree with benthic records. This paragraph was rephrased: "Along with other CIE estimates from well-preserved foraminifera such as in the South Atlantic ($-3.4 \pm 0.2\%$, paleo-water depth of ~1500 m; Supplementary Fig. 1)(McCarren et al., 2008) and in the North Atlantic ($-3.3 \pm 0.2\%$; Supplementary Fig. 1)(Cramer and Kent, 2005b), these estimates are consistent with our $\delta^{13}\text{C}_{\text{Cren}}$ -based CIE_{DIC} estimate, emphasizing that the global median CIE of planktic foraminiferal records (McInerney and Wing, 2011b) of -2.6% is biased by generally poor carbonate preservation. ..."* (Line 308-313)

Line 256: This should be the CIE for the exogenic carbon. It is also stated several times, sometimes without reference. The concept is by no means new.

Following the recommendation, we rephrased this statement to "*CIE of the exogenic carbon reservoir*" and included references to (Dickens et al., 1995) and (Dickens, 2001).

Line 261: This needs to be amended as it is the entire exogenic carbon cycle that needs to be perturbed, not just the ocean. I think the referencing numbers somehow muddled here as the overall concept should be 59.

Following the reviewer's recommendation, we have rephrased this to "*exogenic carbon reservoir*". The reference to Dickens et al. (2001) was added also to the initial statement of the paragraph.

Line 274: As above, I would fix the numbers and clarify the mechanisms (e.g., it is release and oxidation of biogenic methane from the seafloor, oxidation of organic carbon, etc.)

We have rephrased this section to clarify the different scenarios and have revised the mass balance estimates.

Line 277: But of course, as stated for over 20 years, such a mechanism cannot explain all the warming ... as methane release from the seafloor needs to be driven in part by warming. (The same is true with oxidation of organic carbon on land).

Line 278: What is accepted climate sensitivity when this is a highly debated topic?

We agree with the reviewer. However, following the restructuring of this section based on other reviewer comments, these statements were not retained in the revised manuscript.

Line 284: avoid latin abbreviations in direct prose

Removed.

Line 301: Note that Sluijs et al. (2007) measured the d13C on apertodinium so the same phase was used and that bioturbation cannot separate d13C and d18O signals measured on the same phase.

We have rephrased this statement for clarity: *“This approach avoids biases from diagenetic carbonate overgrowth and dissolution affecting foraminiferal $\delta^{13}\text{C}$ (Kozdon et al., 2018b) and stratigraphic ambiguities that arise when separate proxies are used for the CIE and warming.”* (Line 372-374). We politely disagree that temperature and d13C were measured on the same substrate by Sluijs et al. (2007), as they measured temperature (via TEX86) and d13C on distinct substrates, i.e., dinoflagellate cysts versus sedimentary lipids, which may be affected distinctly by differential transport and reworking.

Line 355: size?

Sample size was 10-20 cm³. This has been clarified in the revised text.

References

- Berg I. A., Kockelkorn D., Buckel W. and Fuchs G. (2007) A 3-Hydroxypropionate/4-Hydroxybutyrate Autotrophic Carbon Dioxide Assimilation Pathway in Archaea. *Science* **318**, 1782–1786.
- Church M. J., Wai B., Karl D. M. and DeLong E. F. (2010) Abundances of crenarchaeal *amoA* genes and transcripts in the Pacific Ocean. *Environmental Microbiology* **12**, 679–688.
- Cramer B. S. and Kent D. V. (2005a) Bolide summer: The Paleocene/Eocene thermal maximum as a response to an extraterrestrial trigger. *Palaeogeography, Palaeoclimatology, Palaeoecology* **224**, 144–166.
- Cramer B. S. and Kent D. V. (2005b) Bolide summer: The Paleocene/Eocene thermal maximum as a response to an extraterrestrial trigger. *Palaeogeography, Palaeoclimatology, Palaeoecology* **224**, 144–166.
- Cui Y. and Schubert B. A. (2017) Atmospheric $p\text{CO}_2$ reconstructed across five early Eocene global warming events. *Earth and Planetary Science Letters* **478**, 225–233.
- DeConto R. M., Galeotti S., Pagani M., Tracy D., Schaefer K., Zhang T., Pollard D. and Beerling D. J. (2012) Past extreme warming events linked to massive carbon release from thawing permafrost. *Nature* **484**, 87–91.
- Dickens G. R. (2011) Down the Rabbit Hole: Toward appropriate discussion of methane release from gas hydrate systems during the Paleocene-Eocene thermal maximum and other past hyperthermal events. *Climate of the Past* **7**, 831–846.
- Dickens G. R. (2001) Modeling the Global Carbon Cycle with a Gas Hydrate Capacitor: Significance for the Latest Paleocene Thermal Maximum. In *Geophysical Monograph Series* (eds. C. K. Paull and W. P. Dillon). American Geophysical Union, Washington, D. C. pp. 19–38. Available at: <http://doi.wiley.com/10.1029/GM124p0019> [Accessed October 9, 2018].
- Dickens G. R. (2003) Rethinking the global carbon cycle with a large, dynamic and microbially mediated gas hydrate capacitor. *Earth and Planetary Science Letters* **213**, 169–183.
- Dickens G. R., O'Neil J. R., Rea D. K. and Owen R. M. (1995) Dissociation of oceanic methane hydrate as a cause of the carbon isotope excursion at the end of the Paleocene. *Paleoceanography* **10**, 965–971.
- Dunkley Jones T., Lunt D. J., Schmidt D. N., Ridgwell A., Sluijs A., Valdes P. J. and Maslin M. (2013) Climate model and proxy data constraints on ocean warming across the Paleocene–Eocene Thermal Maximum. *Earth-Science Reviews* **125**, 123–145.
- Frieling J., Peterse F., Lunt D. J., Bohaty S. M., Damsté J. S. S., Reichart G.-J. and Sluijs A. (2019) Widespread warming before and elevated barium burial during the Paleocene-Eocene Thermal Maximum: evidence for methane hydrate release? *Paleoceanography and Paleoclimatology* **34**. Available at: <http://agupubs.onlinelibrary.wiley.com/doi/abs/10.1029/2018PA003425> [Accessed March 20, 2019].

- Gutjahr M., Ridgwell A., Sexton P. F., Anagnostou E., Pearson P. N., Pälike H., Norris R. D., Thomas E. and Foster G. L. (2017) Very large release of mostly volcanic carbon during the Palaeocene–Eocene Thermal Maximum. *Nature* **548**, 573–577.
- Higgins J. A. and Schrag D. P. (2006) Beyond methane: Towards a theory for the Paleocene-Eocene Thermal Maximum. *Earth and Planetary Science Letters* **245**, 523–537.
- Hollis C. J., Taylor K. W. R., Handley L., Pancost R. D., Huber M., Creech J. B., Hines B. R., Crouch E. M., Morgans H. E. G., Crampton J. S., Gibbs S., Pearson P. N. and Zachos J. C. (2012) Early Paleogene temperature history of the Southwest Pacific Ocean: Reconciling proxies and models. *Earth and Planetary Science Letters* **349–350**, 53–66.
- Hopmans E. C., Weijers J. W. H., Schefuß E., Herfort L., Sinninghe Damsté J. S. and Schouten S. (2004) A novel proxy for terrestrial organic matter in sediments based on branched and isoprenoid tetraether lipids. *Earth and Planetary Science Letters* **224**, 107–116.
- Hurley S. J., Close H. G., Elling F. J., Jasper C. E., Gospodinova K., McNichol A. P. and Pearson A. (2019) CO₂-dependent carbon isotope fractionation in Archaea, Part II: The marine water column. *Geochimica et Cosmochimica Acta*.
- John C. M., Bohaty S. M., Zachos J. C., Sluijs A., Gibbs S., Brinkhuis H. and Bralower T. J. (2008) North American continental margin records of the Paleocene-Eocene thermal maximum: Implications for global carbon and hydrological cycling. *Paleoceanography* **23**, PA2217.
- Kirtland Turner S., Hull P. M., Kump L. R. and Ridgwell A. (2017) A probabilistic assessment of the rapidity of PETM onset. *Nature Communications* **8**. Available at: <http://www.nature.com/articles/s41467-017-00292-2> [Accessed October 9, 2018].
- Kirtland Turner S. and Ridgwell A. (2016) Development of a novel empirical framework for interpreting geological carbon isotope excursions, with implications for the rate of carbon injection across the PETM. *Earth and Planetary Science Letters* **435**, 1–13.
- Kohfeld K. E., Anderson R. F. and Lynch-Stieglitz J. (2000) Carbon isotopic disequilibrium in polar planktonic foraminifera and its impact on modern and Last Glacial Maximum reconstructions. *Paleoceanography* **15**, 53–64.
- Kozdon R., Kelly D. C. and Valley J. W. (2018a) Diagenetic Attenuation of Carbon Isotope Excursion Recorded by Planktic Foraminifers During the Paleocene-Eocene Thermal Maximum. *Paleoceanography and Paleoclimatology* **33**, 367–380.
- Kozdon R., Kelly D. C. and Valley J. W. (2018b) Diagenetic Attenuation of Carbon Isotope Excursion Recorded by Planktic Foraminifers During the Paleocene-Eocene Thermal Maximum. *Paleoceanography and Paleoclimatology* **33**, 2017PA003314.
- Kump L. R. and Arthur M. A. (1999) Interpreting carbon-isotope excursions: carbonates and organic matter. *Chemical Geology* **161**, 181–198.

- Laws E. A., Popp B. N., Bidigare J. R. R., Kennicutt M. C. and Macko S. A. (1995) Dependence of phytoplankton carbon isotopic composition on growth rate and $[\text{CO}_2]_{\text{aq}}$: Theoretical considerations and experimental results.
- McCarren H., Thomas E., Hasegawa T., Röhl U. and Zachos J. C. (2008) Depth dependency of the Paleocene-Eocene carbon isotope excursion: Paired benthic and terrestrial biomarker records (Ocean Drilling Program Leg 208, Walvis Ridge). *Geochemistry, Geophysics, Geosystems* **9**, Q10008.
- McInerney F. A. and Wing S. L. (2011a) The Paleocene-Eocene Thermal Maximum: A Perturbation of Carbon Cycle, Climate, and Biosphere with Implications for the Future. *Annual Review of Earth and Planetary Sciences* **39**, 489–516.
- McInerney F. A. and Wing S. L. (2011b) The Paleocene-Eocene Thermal Maximum: A Perturbation of Carbon Cycle, Climate, and Biosphere with Implications for the Future. *Annual Review of Earth and Planetary Sciences* **39**, 489–516.
- Mudelsee M. (2000) Ramp function regression: a tool for quantifying climate transitions. *Computers & Geosciences* **26**, 293–307.
- Pagani M., Pedentchouk N., Huber M., Sluijs A., Schouten S., Brinkhuis H., Sinninghe Damsté J. S., Dickens G. R. and Expedition 302 Scientists (2006) Arctic hydrology during global warming at the Palaeocene/Eocene thermal maximum. *Nature* **442**, 671–675.
- Panchuk K., Ridgwell A. and Kump L. R. (2008) Sedimentary response to Paleocene-Eocene thermal maximum carbon release: A model-data comparison. *Geology* **36**, 315–318.
- Pearson A., Hurley S. J., Elling F. J. and Wilkes E. B. (2019) CO_2 -dependent carbon isotope fractionation in Archaea, Part I: Modeling the 3HP/4HB pathway. *Geochimica et Cosmochimica Acta*.
- Pearson A., Hurley S. J., Shah Walter S. R., Kusch S., Lichtin S. and Zhang Y. G. (2016) Stable carbon isotope ratios of intact GDGTs indicate heterogeneous sources to marine sediments. *Geochimica et Cosmochimica Acta* **181**, 18–35.
- Penman D. E., Hönisch B., Zeebe R. E., Thomas E. and Zachos J. C. (2014) Rapid and sustained surface ocean acidification during the Paleocene-Eocene Thermal Maximum. *Paleoceanography* **29**, 357–369.
- Polik C. A., Elling F. J. and Pearson A. (2018) Impacts of Paleoecology on the TEX_{86} Sea Surface Temperature Proxy in the Pliocene-Pleistocene Mediterranean Sea. *Paleoceanography and Paleoclimatology* **33**, 1472–1489.
- Popp B. N., Laws E. A., Bidigare R. R., Dore J. E., Hanson K. L. and Wakeham S. G. (1998) Effect of Phytoplankton Cell Geometry on Carbon Isotopic Fractionation. *Geochimica et Cosmochimica Acta* **62**, 69–77.
- Ravelo A. C. and Fairbanks R. G. (1995) Carbon Isotopic Fractionation in Multiple Species of Planktonic Foraminifera From Core-Tops in the Tropical Atlantic. *Journal of Foraminiferal Research* **25**, 53–74.

- Santoro A. E., Casciotti K. L. and Francis C. A. (2010) Activity, abundance and diversity of nitrifying archaea and bacteria in the central California Current. *Environmental Microbiology* **12**, 1989–2006.
- Schubert B. A. and Jahren A. H. (2013) Reconciliation of marine and terrestrial carbon isotope excursions based on changing atmospheric CO₂ levels. *Nature communications* **4**, 1653.
- Sluijs A., Bijl P. K., Schouten S., Röhl U., Reichart G.-J. and Brinkhuis H. (2011) Southern ocean warming, sea level and hydrological change during the Paleocene-Eocene thermal maximum. *Climate of the Past* **7**, 47–61.
- Sluijs A., Brinkhuis H., Schouten S., Bohaty S. M., John C. M., Zachos J. C., Reichart G.-J., Sinninghe Damsté J. S., Crouch E. M. and Dickens G. R. (2007) Environmental precursors to rapid light carbon injection at the Palaeocene/Eocene boundary. *Nature* **450**, 1218–21.
- Sluijs A. and Dickens G. R. (2012) Assessing offsets between the $\delta^{13}\text{C}$ of sedimentary components and the global exogenic carbon pool across early Paleogene carbon cycle perturbations. *Global Biogeochemical Cycles* **26**, GB4005.
- Sluijs A., van Roij L., Frieling J., Laks J. and Reichart G.-J. (2018) Single-species dinoflagellate cyst carbon isotope ecology across the Paleocene-Eocene Thermal Maximum. *Geology* **46**, 79–82.
- Sluijs A., Schouten S., Pagani M., Woltering M., Brinkhuis H., Damsté J. S. S., Dickens G. R., Huber M., Reichart G. J., Stein R., Matthiessen J., Lourens L. J., Pedentchouk N., Backman J., Moran K., Clemens S., Cronin T., Eynaud F., Gattacceca J., Jakobsson M., Jordan R., Kaminski M., King J., Koc N., Martinez N. C., McInroy D., Moore T. C., O'Regan M., Onodera J., Pälike H., Rea B., Rio D., Sakamoto T., Smith D. C., St John K. E. K., Suto I., Suzuki N., Takahashi K., Watanabe M. and Yamamoto M. (2006) Subtropical Arctic Ocean temperatures during the Palaeocene/Eocene thermal maximum. *Nature* **441**, 610–613.
- Spero H. J., Bijma J., Lea D. W. and Bemis B. E. (1997) Effect of seawater carbonate concentration on foraminiferal carbon and oxygen isotopes. *Nature* **390**, 497–500.
- Spero H. J. and Lea D. W. (1996) Experimental determination of stable isotope variability in *Globigerina bulloides*: implications for paleoceanographic reconstructions. *Marine Micropaleontology* **28**, 231–246.
- Stassen P., Thomas E. and Speijer R. P. (2012) Integrated stratigraphy of the Paleocene-Eocene thermal maximum in the New Jersey Coastal Plain: Toward understanding the effects of global warming in a shelf environment. *Paleoceanography* **27**, PA4210.
- Svensen H., Planke S., Malthes-Sørensen A., Jamtveit B., Myklebust R., Rasmussen Eidem T. and Rey S. S. (2004) Release of methane from a volcanic basin as a mechanism for initial Eocene global warming. *Nature* **429**, 542–545.
- Tripati A. K., Delaney M. L., Zachos J. C., Anderson L. D., Kelly D. C. and Elderfield H. (2003) Tropical sea-surface temperature reconstruction for the early Paleogene using Mg/Ca ratios of planktonic foraminifera. *Paleoceanography* **18**. Available at:

<http://agupubs.onlinelibrary.wiley.com/doi/abs/10.1029/2003PA000937> [Accessed March 27, 2019].

Weijers J. W. H., Wiesenberg G. L. B., Bol R., Hopmans E. C. and Pancost R. D. (2010) Carbon isotopic composition of branched tetraether membrane lipids in soils suggest a rapid turnover and a heterotrophic life style of their source organism(s). *Biogeosciences* **7**, 2959–2973.

Westerhold T., Röhl U. and Laskar J. (2012) Time scale controversy: Accurate orbital calibration of the early Paleogene: TIME SCALE CONTROVERSY. *Geochemistry, Geophysics, Geosystems* **13**, n/a-n/a.

Westerhold T., Röhl U., Wilkens R. H., Gingerich P. D., Clyde W. C., Wing S. L., Bowen G. J. and Kraus M. J. (2018) Synchronizing early Eocene deep-sea and continental records - Cyclostratigraphic age models for the Bighorn Basin Coring Project drill cores. *Climate of the Past* **14**, 303–319.

Wuchter C., Schouten S., Wakeham S. G. and Sinninghe Damsté J. S. (2005) Temporal and spatial variation in tetraether membrane lipids of marine Crenarchaeota in particulate organic matter: Implications for TEX₈₆ paleothermometry. *Paleoceanography* **20**, PA3013.

Zachos J. C., Röhl U., Schellenberg S. A., Sluijs A., Hodell D. A., Kelly D. C., Thomas E., Nicolo M., Raffi I., Lourens L. J., McCarren H. and Kroon D. (2005) Rapid acidification of the ocean during the Paleocene-Eocene thermal maximum. *Science* **308**, 1611–1615.

Zeebe R. E., Zachos J. C. and Dickens G. R. (2009) Carbon dioxide forcing alone insufficient to explain Palaeocene–Eocene Thermal Maximum warming. *Nature Geoscience* **2**, 576–580.

Reviewers' comments:

Reviewer #1 (Remarks to the Author):

Dear Dr. Scott,

I was happy to see a thoroughly revised version of the paper by Elling et al. They pretty much disagree with most of the issues I have raised in my first review. Much of their argumentation, however, is based on additional published work but there is a wide range of underconstrained factors.

I very much like the approach the authors have taken to change the paper, focusing much more on the potential biases inherent to their discussion. As a result, their numbers are actually very different from the first version. This is I think great but also shows that we are dealing with a proxy in development. The authors are exploring a new proxy for $\delta^{13}\text{C}$ -DIC. In comparison, it has probably taken three decades of international research to constrain carbon isotopic biases in foraminifera, which allows the authors to invoke those to explain differences with their findings. As with all new proxies, it is difficult to get some sense of what is realistic because constraints on the new method are very limited. For a large part, the discussion is now devoted to treating potential uncertainties. The conclusion is that none of the potential uncertainties play a role, but for many of these issues, we really do not have the solid constraints to do this.

I think there is one piece of information in the current paper that shows that the $\delta^{13}\text{C}$ -cren does not only depend on global exogenic $\delta^{13}\text{C}$. I did not spot this in my first review, but at New Jersey and in the Tasman Sea, post CIE $\delta^{13}\text{C}$ -cren values are lower than pre-CIE but in the Arctic Ocean this is the other way around. I think this proves that factors other than the carbon isotopic composition of the global exogenic carbon pool play a role in $\delta^{13}\text{C}$ -cren variations. And I think this will be intuitive to a large part of the community.

Having said that, I think that this paper is worth publishing, even just because these records are very cool and promising. However, I think at this stage, it still suffers from overstatement because the uncertainties in the proxy for local $\delta^{13}\text{C}$ -DIC, and the connection between local $\delta^{13}\text{C}$ -DIC change and average global exogenic $\delta^{13}\text{C}$ change during the PETM are underestimated. The research into $\delta^{13}\text{C}$ -cren is exiting and new but it is in its infancy. The easiest way to tackle this, I would say, is state this up-front and be clear about it in the discussion and conclusions. This is what we know (and that overview is now pretty good), this is what we don't know (this is less clear) and so here is one answer that may come from this data but the uncertainties are hard to quantify. For example, we do not really know that there is no species-specific fractionation in Thaumarcheota. It is a valuable and good hypothesis based on the present information (lines 347-350) but I think the authors would agree that this information is quite limited and so next year this might be different. But the assumption is made without acknowledging this uncertainty. That I think needs to be changed throughout the discussion on such potential biases, and the conclusions.

Below I include all my comments chronologically with the manuscript. I hope they are of use to the authors.

Sincerely,

Appy Sluijs

P.S., I don't review anonymously because I think it makes my reviews more thorough and hopefully constructive, but I clearly forgot to sign my first review.

21-22. "enabling refined estimates for PETM emissions scenarios with reduction in concomitant uncertainty". Not very specific; abstract should state how much refinement and reduction in

uncertainty or just the new numbers.

22-24. Coeval is an indicator of time, while this aspect is studied as a function of depth. In the studied cores at the present resolution, the shifts occur at the same depth but considering potential incompleteness or sedimentary condensation of the studied sediment sequences and the present sampling resolution, this does not automatically provide the constraints required to say that the shifts occurred coevally.

Based on Frieling et al. (2019) you would not expect to find the lead of temperature relative to the CIE in these cores with the temporal sampling resolution. The stratigraphic constraints of Frieling et al. are probably better than those in this new study. So the statement seems a bit strange and superfluous.

However, I fundamentally agree with the sentence in lines 736-738. But it seems that other sites need to be studied to do this.

33. The use of the word 'land' here is bit confusing as the rock reservoir is also mostly land and that is not what is intended (I presume biosphere (also marine) and soils).

158 – 159. '...the change in $\delta^{13}\text{C}_{\text{cren}}$ during the PETM is representative of the CIE in the global exogenic carbon reservoir'. This statement remains problematic, as both myself and the other reviewer attempted to explain.

164. 'Results' I presume this was intended as a header. But the next section is rather 'material and methods' or 'approach'

172. High-resolution is a relative statement. Particularly for the lead/lag relations, it is very important to 1) update existing age models for all the sites, and then 2) determine the temporal resolution of the sampling to be included in the text here. It is important to indicate condensation at Site 1172 and potential hiatuses at the New Jersey site (couple papers on completeness of the sequences).

Figure 1. If the approach the authors take is correct then the recovery of the CIE as measured from $\delta^{13}\text{C}_{\text{cren}}$ should also reflect that of the global exogenic carbon pool. This is at least one basic (although internal) test case for the method. I am curious if the authors would agree on this.

The reason why this is important, and I am sorry that I did not notice it in my first review, is that at New Jersey and in the Tasman Sea, post CIE $\delta^{13}\text{C}_{\text{cren}}$ values are lower than pre-CIE but in the Arctic Ocean this is the other way around. I think this proves that factors other than the carbon isotopic composition of the global exogenic carbon pool play a role in $\delta^{13}\text{C}_{\text{cren}}$ variations. How can we tell that such factors (whatever they are) did not play a role at the onset of the CIE at these sites?

282. Concomitant is an indicator of time, while this aspect is studied as a function of depth. In this core at the present resolution, the shifts occur at the same depth but considering potential incompleteness of the core and a certain sampling resolution, this does not automatically provide the constraints required to say that the shifts occurred concomitantly.

292-293. "There is no discernible offset in timing between changes in the TEX86 and $\delta^{13}\text{C}_{\text{cren}}$ signals at the PETM onset". There is in the Arctic record (see my previous review).

337. Does "(~850-2200 ppm)" mean the range of estimates by which the CO_2 concentration rose during the PETM? If so, is the background and maximum value not as important? The authors would probably agree that it is fair to acknowledge that we don't know this parameter very well. Some extremes include the scenario's of Gutjahr et al. [2017] and Zeebe et al., [2009]. Does the uncertainty reported here include that complete range?

347-350. Right now the assumption is taken that this is correct. This would be great and it might be true, but it is untested so we don't really know this. Is it correct that no error is assigned to

species-specific fractionation? If so, I think this needs to be explicitly mentioned as a potential source of uncertainty.

397-418. I don't quite get this argument. Now ref 58 and Zeebe and Zachos [2007] actually suggest circulation changed direction during the PETM. If so, it should have affected d13C-DIC differently in the Pacific and Atlantic Site (ref 58 tries to do this but it remains very difficult because of dissolution). If this would have affected the d13C-cren records, the one CIE should have been larger than the other because the sampled DIC would have been older and younger than before the PETM, respectively.

I think reality saves you here, though because while circulation patterns can be seen in deep ocean d13C-DIC gradients, they cannot be seen in the surface ocean because of equilibration with the atmosphere and several other confounding factors along continental margins, such as the studied sites.

In addition, ref 58 is just one of the studies published on Paleogene circulation but the real interesting issue is the lack of clear gradients between the Indo-Pacific and Indo-Atlantic, which perhaps implies two independent circulation cells [Luo, et al., 2016]. Much discussion on this, hence, and so to use conclusions of one of these papers to make an argument (regardless of the above depth problem) is a stretch.

548-558. I think this text originates from a point I apparently did not make clearly in my first review. Of course the injected carbon mixed with the global exogenic carbon pool in a couple thousand years. However, climate change must have systematically changed the distribution of carbon isotopes within the exogenic carbon pool for the duration of the PETM. For example, there is evidence for increased seasonal precipitation, stratification and sedimentation as well as transport of terrigenous organic matter to continental shelves. Such processes should have offset the change in local d13C-DIC pools from the average change in the global exogenic carbon pool (one of the points Dickens and Sluijs made). And so the CIE in DIC is not expected to be identical everywhere in the ocean, certainly in near-shore environments.

M. Gutjahr, A. Ridgwell, P.F. Sexton, E. Anagnostou, P.N. Pearson, H. Pälike, R.D. Norris, E. Thomas, G.L. Foster, Very large release of mostly volcanic carbon during the Palaeocene-Eocene Thermal Maximum, *Nature* 548(2017) pp. 573.

Y. Luo, B.P. Boudreau, G.R. Dickens, A. Sluijs, J.J. Middelburg, An alternative model for CaCO₃ over-shooting during the PETM: Biological carbonate compensation, *Earth and Planetary Science Letters* 453(2016) pp. 223-233.

R.E. Zeebe, J.C. Zachos, Reversed deep-sea carbonate ion basin gradient during the Paleocene-Eocene thermal maximum, *Paleoceanography* 22(2007) pp. PA3301.

R.E. Zeebe, J.C. Zachos, G.R. Dickens, Carbon dioxide forcing alone insufficient to explain Palaeocene-Eocene Thermal Maximum warming, *Nature Geoscience* 2(2009) pp. 576-580.

Reviewer #2 (Remarks to the Author):

Dear Editor and Authors,

I was fairly critical of the initial submission, but thought that it had a good backbone. I have read the letter of response by the authors to referee comments as well as the revised manuscript. Let me be the first to congratulate the authors in rethinking and rewriting a superior manuscript.

I'll be up front -- I still do not agree with some of the conclusions. However, I very much liked

reading this new revised version, and disagreement and discussion very much drives great science. The cool and novel aspects of the work are now aligned with pushing science forward. I think the manuscript should be published and that Nature Communications is an excellent venue. It is a novel and nice piece of work on a great topic.

I suggest publication almost "as is", but make a few comments below.

Sincerely,

Gerald (Jerry) Dickens

We politely disagree that temperature and $\delta^{13}\text{C}$ were measured on the same substrate by Sluijs et al. (2007), as they measured temperature (via TEX86) and $\delta^{13}\text{C}$ on distinct substrates, i.e., dinoflagellate cysts versus sedimentary lipids, which may be affected distinctly by differential transport and reworking

.

The above is a very good point, one that I had not fully thought about.

Line 322-324: For fun, read Dickens (Bull. Geol. Soc. France, 2000).

Line 364-371: Some of this will not make sense to most readers. For example "too high" in reference to carbon isotope ratios – what does this mean? All should be very specific in terms of enriched in ^{12}C , ^{13}C , etc.

From a philosophical view, having thought about the root problem from many angles for almost 25 years, when everything is all aligned, the exogenic carbon cycle changed across the PETM by nominally -3 per mil and all else represents how carbon isotopes ($\delta^{13}\text{C}$) gets incorporated into various phases. With the current suggestion of closer to -4 per mil, the obvious answer is that I am just wrong, which is definitely a possibility. Note here, though, that no benthic foraminifera $\delta^{13}\text{C}$ record from an open ocean site shows a -4 per mil excursion, and well, much of the deep sea cannot be sampled because below the CCD.

So, assuming and hoping this work gets published, which it should be, my immediate thought is why might archaea amplify the $\delta^{13}\text{C}$ record during an extreme climate event? This is addressed in the present manuscript somewhat, but it would behoove the authors to more fully think about, such that in XX years in the future, all bases are covered and the work remains important.

Reviewer #1 (Remarks to the Author):

Dear Dr. Scott,

I was happy to see a thoroughly revised version of the paper by Elling et al. They pretty much disagree with most of the issues I have raised in my first review. Much of their argumentation, however, is based on additional published work but there is a wide range of underconstrained factors.

I very much like the approach the authors have taken to change the paper, focusing much more on the potential biases inherent to their discussion. As a result, their numbers are actually very different from the first version. This is I think great but also shows that we are dealing with a proxy in development. The authors are exploring a new proxy for $\delta^{13}\text{C-DIC}$. In comparison, it has probably taken three decades of international research to constrain carbon isotopic biases in foraminifera, which allows the authors to invoke those to explain differences with their findings. As with all new proxies, it is difficult to get some sense of what is realistic because constraints on the new method are very limited. For a large part, the discussion is now devoted to treating potential uncertainties. The conclusion is that none of the potential uncertainties play a role, but for many of these issues, we really do not have the solid constraints to do this.

I think there is one piece of information in the current paper that shows that the $\delta^{13}\text{C-cren}$ does not only depend on global exogenic $\delta^{13}\text{C}$. I did not spot this in my first review, but at New Jersey and in the Tasman Sea, post CIE $\delta^{13}\text{C-cren}$ values are lower than pre-CIE but in the Arctic Ocean this is the other way around. I think this proves that factors other than the carbon isotopic composition of the global exogenic carbon pool play a role in $\delta^{13}\text{C-cren}$ variations. And I think this will be intuitive to a large part of the community.

Having said that, I think that this paper is worth publishing, even just because these records are very cool and promising. However, I think at this stage, it still suffers from overstatement because the uncertainties in the proxy for local $\delta^{13}\text{C-DIC}$, and the connection between local $\delta^{13}\text{C-DIC}$ change and average global exogenic $\delta^{13}\text{C}$ change during the PETM are underestimated.

The research into $\delta^{13}\text{C-cren}$ is exciting and new but it is in its infancy. The easiest way to tackle this, I would say, is state this up-front and be clear about it in the discussion and conclusions. This is what we know (and that overview is now pretty good), this is what we don't know (this is less clear) and so here is one answer that may come from this data but the uncertainties are hard to quantify. For example, we do not really know that there is no species-specific fractionation in Thaumarcheota. It is a valuable and good hypothesis based on the present information (lines 347-350) but I think the authors would agree that this information is quite limited and so next year this might be different. But the assumption is made without acknowledging this uncertainty. That I think needs to be changed throughout the discussion on such potential biases, and the conclusions.

Below I include all my comments chronologically with the manuscript. I hope they are of use to the authors.

Sincerely,

Appy Sluijs

P.S., I don't review anonymously because I think it makes my reviews more thorough and hopefully constructive, but I clearly forgot to sign my first review.

We appreciate your thorough reviews, which helped greatly improve our manuscript. We agree that some aspects of the $\delta^{13}\text{C}$ -Cren proxy remain underconstrained and that the Arctic Ocean records may show evidence for local effects on $\delta^{13}\text{C}$ -DIC or $\delta^{13}\text{C}$ -Cren. In our revision, we address these issues and clarify the remaining uncertainties of the $\delta^{13}\text{C}$ -Cren proxy, as described in detail below.

21-22. "enabling refined estimates for PETM emissions scenarios with reduction in concomitant uncertainty". Not very specific; abstract should state how much refinement and reduction in uncertainty or just the new numbers.

We agree and have rephrased this sentence:

Lines 19-23: "Novel records of the isotopic composition of crenarchaeol ($\delta^{13}\text{C}_{\text{cren}}$) constrain the global CIE magnitude to $-4.0 \pm 0.4\%$, consistent with emission of $>3,000$ Pg C from methane hydrate dissociation or $>4,400$ Pg C for scenarios involving emissions from geothermal heating or oxidation of sedimentary organic matter."

22-24. Coeval is an indicator of time, while this aspect is studied as a function of depth. In the studied cores at the present resolution, the shifts occur at the same depth but considering potential incompleteness or sedimentary condensation of the studied sediment sequences and the present sampling resolution, this does not automatically provide the constraints required to say that the shifts occurred coevally.

Based on Frieling et al. (2019) you would not expect to find the lead of temperature relative to the CIE in these cores with the temporal sampling resolution. The stratigraphic constraints of Frieling et al. are probably better than those in this new study. So the statement seems a bit strange and superfluous. However, I fundamentally agree with the sentence in lines 736-738. But it seems that other sites need to be studied to do this.

We agree that the present cores limit determination of lead-lag relationships and have thus removed this statement from the abstract. We have rephrased the respective discussion sentence:

Lines 391-396: "Although incompleteness of the PETM onset in the Arctic Ocean record and potentially the Tasman core limits the determination of lead-lag relationships, the parallel changes in $\delta^{13}\text{C}_{\text{cren}}$ and TEX_{86} (Fig. 2) suggests that any warming preceding the CIE must have occurred no more than 3-4 ka prior, consistent with previous estimates from the New Jersey shelf^{38,76}, the equatorial Atlantic⁷⁶, and the Southern Ocean⁷⁶."

33. The use of the word 'land' here is bit confusing as the rock reservoir is also mostly land and that is not what is intended (I presume biosphere (also marine) and soils).

Changed to "biosphere".

158 – 159. ‘...the change in $\delta^{13}\text{C}_{\text{cren}}$ during the PETM is representative of the CIE in the global exogenic carbon reservoir’. This statement remains problematic, as both myself and the other reviewer attempted to explain.

We agree that this statement requires more nuance. We rephrased the sentence to more closely reflect the language used in our discussion:

Lines 91-92: “...that the change in $\delta^{13}\text{C}_{\text{cren}}$ during the PETM closely approximates the CIE in the global exogenic carbon reservoir.”

Based on additional reviewer comments below, we further included statements on the remaining uncertainties of $\delta^{13}\text{C}_{\text{cren}}$ records and the influence of local offsets relative to the global CIE on these records:

Lines 262-265: “Because $\delta^{13}\text{C}_{\text{DIC}}$ dynamics may differ between shelf environments (New Jersey, Tasman Sea), enclosed basins (Arctic Ocean), and deep open ocean environments, additional records from deep ocean environments could help constrain the influence of local effects on $\delta^{13}\text{C}_{\text{DIC}}$ and $\delta^{13}\text{C}_{\text{cren}}$.”

Lines 248-262: “The close agreement in CIE estimates at all sites suggests no significant influence of changes in local environmental conditions on CIE expression. Still, comparatively low $\delta^{13}\text{C}_{\text{cren}}$ values in the Arctic Ocean, when compared to the Tasman Sea and New Jersey shelf, indicate distinct environmental conditions for the Arctic basin... Further, post-PETM $\delta^{13}\text{C}_{\text{cren}}$ values in the Arctic Ocean are equal to or higher than pre-PETM $\delta^{13}\text{C}_{\text{cren}}$ values, while they are lower than pre-PETM values in the Tasman Sea and New Jersey records (Fig. 1). This suggests that during the recovery phase the Arctic Ocean entered a new ecosystem state with a distinct local $\delta^{13}\text{C}_{\text{DIC}}$ signal. ... However, it remains unconstrained how these ecosystem changes, potentially resulting from variability in ocean stratification, affected local $\delta^{13}\text{C}_{\text{DIC}}$ values.”

Lines 418-421: “Further ground-truthing using a wider range of thaumarchaeal cultures and globally representative environmental datasets will help further constrain uncertainties of the $\delta^{13}\text{C}_{\text{cren}}$ proxy and will strengthen its application to the paleoenvironment.”

164. ‘Results’ I presume this was intended as a header. But the next section is rather ‘material and methods’ or ‘approach’

We have added a header to this subsection, “Study area”.

172. High-resolution is a relative statement. Particularly for the lead/lag relations, it is very important to 1) update existing age models for all the sites, and then 2) determine the temporal resolution of the sampling to be included in the text here. It is important to indicate condensation at Site 1172 and potential hiatuses at the New Jersey site (couple papers on completeness of the sequences).

We agree and have deleted “high-resolution”. We did not find literature evidence for condensation or basal unconformities in the Ancora record. Rather, Stassen et al. suggest that the Ancora record is not or less impacted by condensation compared to other New Jersey records where lead-lag relationships have been determined. Evidence for condensation in the Tasman Sea record is limited but we agree that condensation is plausible given the low carbonate contents. We agree that further studies from cores with better age constraints are

needed and state this in the revised manuscript. As further stated in the manuscript, our results are consistent with literature data despite the valid concerns raised by the reviewer. We have modified the respective discussion to reflect this view:

Lines 391-396: “Although incompleteness of the PETM onset in the Arctic Ocean record, and potentially the Tasman core, as well as a lack of well-constrained age models limits the determination of lead-lag relationships, the parallel changes in $\delta^{13}\text{C}_{\text{Cren}}$ and TEX_{86} (Fig. 2) suggests that any warming preceding the CIE must have occurred no more than 3-4 ka prior, consistent with previous estimates from the New Jersey shelf^{38,76}, the equatorial Atlantic⁷⁶, and the Southern Ocean⁷⁶.”

Figure 1. If the approach the authors take is correct then the recovery of the CIE as measured from $\delta^{13}\text{C}_{\text{Cren}}$ should also reflect that of the global exogenic carbon pool. This is at least one basic (although internal) test case for the method. I am curious if the authors would agree on this.

The reason why this is important, and I am sorry that I did not notice it in my first review, is that at New Jersey and in the Tasman Sea, post-CIE $\delta^{13}\text{C}_{\text{Cren}}$ values are lower than pre-CIE but in the Arctic Ocean this is the other way around. I think this proves that factors other than the carbon isotopic composition of the global exogenic carbon pool play a role in $\delta^{13}\text{C}_{\text{Cren}}$ variations. How can we tell that such factors (whatever they are) did not play a role at the onset of the CIE at these sites?

We agree that the recovery in each record should be representative of the global CIE or otherwise reveal local offsets. Although about half of the post-PETM $\delta^{13}\text{C}_{\text{Cren}}$ values in the Arctic Ocean are equal to pre-PETM values, some values are higher, and none are lower than the pre-PETM. We agree that this distinct record reveals local effects on $\delta^{13}\text{C}_{\text{Cren}}$. Although we cannot identify the source of this change, it appears that unlike in the Tasman Sea and New Jersey shelf, Arctic Ocean biogeochemistry did not recover to original conditions after the PETM but instead entered a new state. This can be readily observed in bulk sediment nitrogen isotopes (Knies et al., 2008), which are sensitive indicators of both the nitrogen cycle processes as well as water column redox state. Bulk sediment nitrogen isotopes are ~ 3 permil before the PETM, ~ 0.5 permil during the PETM, and ~ 2 permil after the PETM (Knies et al., 2008). This indicates that due to its enclosed setting the Arctic Ocean remained poorly ventilated and potentially less ventilated than before the PETM. The cause for this behavior and its influence on seawater $\delta^{13}\text{C}_{\text{Cren}}$ remains unknown. However, this would not affect our pre-PETM to body PETM $\delta^{13}\text{C}_{\text{Cren}}$ estimates. We think that with a more complete record of the Arctic Ocean and records from additional environments (e.g., deep open ocean), the effects of local changes could be more thoroughly assessed. To address this issue, we have added a paragraph to the discussion:

Lines 248-262: “The close agreement in CIE estimates at all sites suggests no significant influence of changes in local environmental conditions on CIE expression. Still, comparatively low $\delta^{13}\text{C}_{\text{Cren}}$ values in the Arctic Ocean, when compared to the Tasman Sea and New Jersey shelf, indicate distinct environmental conditions for the Arctic basin. Lower absolute $\delta^{13}\text{C}_{\text{Cren}}$ values may reflect accumulation of ^{13}C -depleted DIC in the chemocline and anoxic zone of the Arctic Ocean due to reduced mixing under enclosed, euxinic conditions similar to the modern Black Sea or Cariaco Basin⁶⁴. Further, post-PETM $\delta^{13}\text{C}_{\text{Cren}}$ values in the Arctic Ocean are equal to or higher than pre-PETM $\delta^{13}\text{C}_{\text{Cren}}$ values, while they are lower than pre-PETM values in the Tasman

Sea and New Jersey records (Fig. 1). This suggests that during the recovery phase the Arctic Ocean entered a new ecosystem state with a distinct local $\delta^{13}\text{C}_{\text{DIC}}$ signal. Bulk sediment $\delta^{15}\text{N}$ records⁶⁵ support the notion that Arctic Ocean ecosystem changes persisted past the PETM. Specifically, lower $\delta^{15}\text{N}$ values in the PETM recovery phase than during the pre-PETM suggest continued suboxic conditions that were more pronounced than prior to the PETM. However, it remains unconstrained how these ecosystem changes, potentially resulting from variability in ocean stratification, affected local $\delta^{13}\text{C}_{\text{DIC}}$ values.”

282. Concomitant is an indicator of time, while this aspect is studied as a function of depth. In this core at the present resolution, the shifts occur at the same depth but considering potential incompleteness of the core and a certain sampling resolution, this does not automatically provide the constraints required to say that the shifts occurred concomitantly.

We agree with the reviewer and have changed the wording to refer to depth rather than time: “At the depth of the CIE...”

292-293. “There is no discernible offset in timing between changes in the TEX₈₆ and $\delta^{13}\text{C}_{\text{Cren}}$ signals at the PETM onset”. There is in the Arctic record (see my previous review).

We performed cross-correlation analyses on the TEX₈₆’ and d13C-Cren data (pre-PETM and onset data only) and found a lag of TEX₈₆’ relative to d13C-Cren in the Arctic record and no lead/lag in the other records. However, we agree that due to the stratigraphic uncertainty of PETM onset in the Arctic record lead/lag relationships remain ambiguous. We did not find literature evidence for basal unconformities or condensation at Ancora (as mentioned above); rather it seems that condensation at Ancora is not or less of an issue compared to other New Jersey sites such as Bass River (Stassen et al., 2012). In order to acknowledge potential biases, we have rephrased the sentence to:

Lines 164-167: “There is no consistent depth offset between changes in the TEX₈₆ (or TEX₈₆’) and $\delta^{13}\text{C}_{\text{Cren}}$ signals at the PETM onset, potentially indicating synchronous change in temperature and CIE. However, incomplete recovery in the Arctic Ocean record^{39,43} and potential condensation in the Tasman Sea record⁴⁷ could have obscured potential lead-lag relationships.”.

337. Does “(~850-2200 ppm)” mean the range of estimates by which the CO₂ concentration rose during the PETM? If so, is the background and maximum value not as important? The authors would probably agree that it is fair to acknowledge that we don’t know this parameter very well. Some extremes include the scenario’s of Gutjahr et al. [2017] and Zeebe et al., [2009]. Does the uncertainty reported here include that complete range?

We agree that Paleocene-Eocene pCO₂ is currently not well constrained. We therefore chose to accommodate a wide range of estimates (850-2200 ppm), which derives from model results of the Gutjahr et al. and Zeebe et al. papers. This range does not reflect the rise in pCO₂ but the range of expected values during the latest Paleocene and the PETM. The uncertainty in pre and peak PETM pCO₂ levels is propagated into the d13C-Cren-based CIE estimate as outlined in lines 472-483. We have rephrased and expanded this section to clarify these issues:

Lines 184-190: “Due to the pCO₂ increase during the PETM $\epsilon_{\text{DIC-Cren}}$ may have been smaller by $0.75 \pm 0.15\text{‰}$ (range 0.6-0.9‰) during the peak-PETM compared to the pre-PETM (assuming

Paleocene-Eocene atmospheric $p\text{CO}_2$ of $\sim 850\text{--}2200$ ppm, refs. ^{13,17}; Supplementary Fig. 4). Accounting for this effect leads to a $\delta^{13}\text{C}_{\text{Cren}}$ -based CIE in DIC (CIE_{DIC}) of $-3.87 \pm 0.25\text{‰}$ at Ancora, $-4.03 \pm 0.27\text{‰}$ in the Tasman Sea, $-4.06 \pm 0.27\text{‰}$ in the Arctic Ocean, yielding an average of $-4.0 \pm 0.4\text{‰}$. Tighter constraints on Paleocene-Eocene $p\text{CO}_2$ will allow reducing the uncertainty in $\epsilon_{\text{DIC-Cren}}$ estimates.”

347-350. Right now the assumption is taken that this is correct. This would be great and it might be true, but it is untested so we don't really know this. Is it correct that no error is assigned to species-specific fractionation? If so, I think this needs to be explicitly mentioned as a potential source of uncertainty.

Currently there is no error assigned to species-specific fractionation. Our recently published environmental data line up with our theoretical prediction and imply no or only minimal species-specific effects (Hurley et al., 2019). However, additional studies from other ocean basins and additional cultures will be needed to fully constrain this effect. Following the recommendation of the reviewer, we have added two statements (early and at the end of the discussion) regarding this uncertainty:

Lines 196-197: “Studies with additional thaumarchaeal cultures will be needed to fully assess potential impacts of changes in community composition on $\epsilon_{\text{DIC-Cren}}$ and $\delta^{13}\text{C}_{\text{Cren}}$ records.”

Lines 418-421: “Further ground-truthing using a wider range of thaumarchaeal cultures and globally representative environmental datasets will help quantify uncertainties of the $\delta^{13}\text{C}_{\text{Cren}}$ proxy and will strengthen its application to the paleoenvironment.”

397-418. I don't quite get this argument. Now ref 58 and Zeebe and Zachos [2007] actually suggest circulation changed direction during the PETM. If so, it should have affected $\delta^{13}\text{C-DIC}$ differently in the Pacific and Atlantic Site (ref 58 tries to do this but it remains very difficult because of dissolution). If this would have affected the $\delta^{13}\text{C-Cren}$ records, the one CIE should have been larger than the other because the sampled DIC would have been older and younger than before the PETM, respectively.

I think reality saves you here, though because while circulation patterns can be seen in deep ocean $\delta^{13}\text{C-DIC}$ gradients, they cannot be seen in the surface ocean because of equilibration with the atmosphere and several other confounding factors along continental margins, such as the studied sites.

In addition, ref 58 is just one of the studies published on Paleogene circulation but the real interesting issue is the lack of clear gradients between the Indo-Pacific and Indo-Atlantic, which perhaps implies two independent circulation cells [Luo, et al., 2016]. Much discussion on this, hence, and so to use conclusions of one of these papers to make an argument (regardless of the above depth problem) is a stretch.

We agree that this issue was not properly presented in the previous version. We have added further citations and rephrased this paragraph to reflect the ongoing debate and complexity surrounding ocean circulation changes during the PETM:

Lines 239-243: “... Although lower $\delta^{13}\text{C}_{\text{Cren}}$ values in the northern Atlantic and Arctic compared to the Pacific Ocean support previous arguments for distinct ocean circulation patterns during the

Paleocene-Eocene⁶⁰⁻⁶³, similar CIE magnitudes at all sites suggest that the effect of potential deep water circulation reversals on the CIE was minor in comparison to whole-ocean $\delta^{13}\text{C}_{\text{DIC}}$ changes. ...”

548-558. I think this text originates from a point I apparently did not make clearly in my first review. Of course the injected carbon mixed with the global exogenic carbon pool in a couple thousand years. However, climate change must have systematically changed the distribution of carbon isotopes within the exogenic carbon pool for the duration of the PETM. For example, there is evidence for increased seasonal precipitation, stratification and sedimentation as well as transport of terrigenous organic matter to continental shelves. Such processes should have offset the change in local $\delta^{13}\text{C}$ -DIC pools from the average change in the global exogenic carbon pool (one of the points Dickens and Sluijs made). And so the CIE in DIC is not expected to be identical everywhere in the ocean, certainly in near-shore environments.

We agree that local effects could have influenced $\delta^{13}\text{C}$ -DIC values recorded by $\delta^{13}\text{C}$ -Cren. The absolute $\delta^{13}\text{C}$ -DIC values resulting from our $\delta^{13}\text{C}$ -Cren data were different across our study sites, which may be partially attributed to local factors. However, it remains unconstrained whether these local factors changed significantly across the PETM (in particular relative to magnitude of the CIE itself) and whether these factors changed distinctly or similarly at each study site. The agreement between the three CIEs suggests that divergence in local effects was small relative to the global CIE and thus leads us to propose that the CIE in $\delta^{13}\text{C}$ -Cren approximates the global CIE. We agree that local effects could still be important in shaping $\delta^{13}\text{C}$ -Cren records because $\delta^{13}\text{C}$ -DIC dynamics may be different in shelf environments (New Jersey, Tasman Sea) relative to enclosed basins (Arctic Ocean) or deep open ocean environments. Future studies of $\delta^{13}\text{C}$ -Cren records from deep ocean environments could help constrain the influence of local effects, as now stated in lines 262-265:

“Because $\delta^{13}\text{C}_{\text{DIC}}$ dynamics may differ between shelf environments (New Jersey, Tasman Sea), enclosed basins (Arctic Ocean), and deep open ocean environments, additional records from deep ocean environments could help constrain the influence of local effects on $\delta^{13}\text{C}_{\text{DIC}}$ and $\delta^{13}\text{C}_{\text{Cren}}$.”

Further, we have rephrased an earlier paragraph in the discussion to address the potential influence of local factors and have highlighted and discussed the case of the Arctic Ocean (lines 248-274), as also explained in detail above in response to the comment on Figure 1.

Reviewer #2 (Remarks to the Author):

Dear Editor and Authors,

I was fairly critical of the initial submission, but thought that it had a good backbone. I have read the letter of response by the authors to referee comments as well as the revised manuscript. Let me be the first to congratulate the authors in rethinking and rewriting a superior manuscript.

I'll be up front -- I still do not agree with some of the conclusions. However, I very much liked reading this new revised version, and disagreement and discussion very much drives great science. The cool and novel aspects of the work are now aligned with pushing science forward. I think the manuscript should be published and that Nature Communications is an excellent venue. It is a novel and nice piece of work on a great topic.

I suggest publication almost "as is", but make a few comments below.

Sincerely,

Gerald (Jerry) Dickens

Thank you for your positive assessment and your comments, which helped greatly improve our manuscript. In the present revision, we further emphasize the remaining uncertainties of the d13C-Cren proxy.

We politely disagree that temperature and d13C were measured on the same substrate by Sluijs et al. (2007), as they measured temperature (via TEX86) and d13C on distinct substrates, i.e., dinoflagellate cysts versus sedimentary lipids, which may be affected distinctly by differential transport and reworking

The above is a very good point, one that I had not fully thought about.

Line 322-324: For fun, read Dickens (Bull. Geol. Soc. France, 2000).

Thank you for the recommendation. We have read this paper with great interest.

Line 364-371: Some of this will not make sense to most readers. For example "too high" in reference to carbon isotope ratios – what does this mean? All should be very specific in terms of enriched in 12C, 13C, etc.

We have clarified the terminology by rephrasing this sentence to:

Lines 380: "pointed to carbon source $\delta^{13}\text{C}$ values that are too enriched in ^{13}C "

From a philosophical view, having thought about the root problem from many angles for almost 25 years, when everything is all aligned, the exogenic carbon cycle changed across the PETM by nominally -3 per mil and all else represents how carbon isotopes (d13C) gets incorporated into various phases. With the current suggestion of closer to -4 per mil, the obvious answer is that I am just wrong, which is definitely a possibility. Note here, though, that no benthic foraminifera d13C record from an open ocean site shows a -4 per mil excursion, and well, much of the deep sea cannot be sampled because below the CCD.

So, assuming and hoping this work gets published, which it should be, my immediate thought is why might archaea amplify the d13C record during an extreme climate event? This is addressed in the

present manuscript somewhat, but it would behoove the authors to more fully think about, such that in XX years in the future, all bases are covered and the work remains important.

In this revised version of the manuscript, we have included assessments of all potential biases of $\delta^{13}\text{C}$ -Cren in the manuscript to the best of our knowledge. We have emphasized the remaining sources of uncertainty in ^{13}C -cren records, for example changes in growth rates or species-specific ^{13}C -offsets. We are currently working on tighter constraints on archaeal $\delta^{13}\text{C}$ records, using both culture experiments and environmental samples. We agree with the reviewers that this proxy is in an early stage of development and believe that this study will inspire follow-up work by the wider community.